# The role of cyclonic activity in tropical temperature-rainfall scaling

Dominik Traxl [1,2✉], Niklas Boers [2,3,4], Aljoscha Rheinwalt [1] & Bodo Bookhagen [1]

The attribution of changing intensity of rainfall extremes to global warming is a key challenge of climate research. From a thermodynamic perspective, via the Clausius-Clapeyron relationship, rainfall events are expected to become stronger due to the increased water-holding capacity of a warmer atmosphere. Here, we employ global, 1-hourly temperature and 3-hourly rainfall data to investigate the scaling between temperature and extreme rainfall. Although the Clausius-Clapeyron scaling of $+7\%$ rainfall intensity increase per degree warming roughly holds on a global average, we find very heterogeneous spatial patterns. Over tropical oceans, we reveal areas with consistently strong negative scaling (below $-40\%\,^{\circ}C^{-1}$). We show that the negative scaling is due to a robust linear correlation between pre-rainfall cooling of near-surface air temperature and extreme rainfall intensity. We explain this correlation by atmospheric and oceanic dynamics associated with cyclonic activity. Our results emphasize that thermodynamic arguments alone are not enough to attribute changing rainfall extremes to global warming. Circulation dynamics must also be thoroughly considered.

---

[1] Institute of Earth and Environmental Science, University of Potsdam, Potsdam, Germany. [2] Potsdam Institute for Climate Impact Research, Potsdam, Germany. [3] Technical University of Munich, School of Engineering & Design, Earth System Modelling, Munich, Germany. [4] Global Systems Institute and Department of Mathematics, University of Exeter, Exeter, UK. ✉email: dominik.traxl@posteo.org

Extreme rainfall can lead to severe natural disasters such as flash floods and landslides. Such events threaten lives, infrastructure, and natural ecosystems. This poses a major threat to our socioeconomic welfare[1,2]. The highest property damages of weather extreme events are typically caused by tropical cyclones (TCs), making them an object of great interest to researchers, disaster preparedness organizations, and also the insurance industry[3]. Therefore, it is vital to understand how extreme rainfall will evolve under current and future anthropogenic global warming[4–7].

Long-term historical rainfall data are scarce[8–10], particularly on subdaily time scales[11–13]. It is, nevertheless, generally argued that extreme rainfall will intensify as our climate warms[14,15]. This is based on the thermodynamic argument that the atmospheric moisture-holding capacity increases with temperature at an exponential rate (~7%°C$^{-1}$), as inferred from the Clausius-Clapeyron (CC) relation[16]. This thermodynamic relationship has been widely used as a benchmark to interpret changes in extreme rainfall due to changes in air temperature[4,17–22]. Indeed, several studies have found an approximate 7%°C$^{-1}$ increase in rainfall rates on a global average, using various analysis methods and temperature covariates[18,23,24].

However, in observed and simulated data, spatial and seasonal deviations from the CC relationship for the temperature-rainfall scaling have been found over many parts of the globe[25]. A number of factors play a role in this context. For example, for rainfall to scale at the CC rate, relative humidity must remain constant[17,26,27]. In high-temperature regimes (e.g. above 24 °C) such as the (sub-)tropics, negative scaling rates have been reported repeatedly, indicating a decrease in rainfall intensity with warming air temperatures[27–30], seemingly due to limited relative humidity in the atmosphere[22,26,27,31,32]. This finding, however, is inconsistent with studies showing a rise in the intensity and frequency of extreme rainfall in past observations[9,33] and future projections[34,35] over most of the globe. Another important factor to consider when analysing the temperature-intensity scaling is that rainfall itself has a cooling effect on the surface air temperature, with higher intensities resulting in stronger cooling[36–38]. Arguably the most important factor, however, is that at regional and local scales, circulation-dynamic responses (e.g. TCs or mesoscale convective systems) can play a pivotal role in the scaling. This makes robust quantifications very challenging[39–42].

In this study, we perform a (nearly) global analysis of the relationship between temperature and extreme-rainfall intensity. We define extreme rainfall as rainfall events above the 90th percentile of wet times, i.e. 3 hourly rainfall events with average rainfall rates above 0.1 mmh$^{-1}$. We employ gridded 3-hourly rainfall data from the Tropical Rainfall Measurement Mission (TRMM[43]) and 1-hourly surface-temperature data from the ERA5 reanalysis data[44] at a spatial resolution of 0.25°. We make sure to mitigate against effects that could influence the temperature-rainfall scaling relation in unintended ways, including: surface cooling by the rainfall events themselves and effects related to the diurnal and seasonal cycles. We focus on tropical oceans to quantitatively describe the influence of the oceanic and atmospheric dynamics associated with cyclones on the apparent temperature-rainfall scaling.

## Results

The following results are based on rainfall events in July–August–September–October (JASO), the TC season of the northern hemisphere. Corresponding results for December–January–February–March–April (DJFMA)—the TC season of the southern hemisphere—can be found in the supplementary material.

**Global temperature-rainfall scaling factors**. We applied an exponential regression between temperature and extreme-rainfall intensity for each grid cell of the data covering the globe from 50°S to 50°N; the resulting scaling factors ($\alpha$-values, in units of %°C$^{-1}$, see Methods) are depicted in Fig. 1a and Supplementary Fig. 1. With a value of 6.0%°C$^{-1}$, the global median of all $\alpha$-values (considering only locations with sufficient data points, see caption of Fig. 1a) is close to the thermodynamic CC relationship. However, pronounced spatial variations are apparent, particularly when comparing $\alpha$-values over water bodies (Fig. 1a) with those over landmasses (Supplementary Fig. 1). Much stronger deviations from the 7% scaling are observed over the oceans, with both positive and negative $\alpha$-values (note the differences in colormap ranges). Approximately 33% of all ocean locations have a negative $\alpha$ and about 53% show an $\alpha$ larger than 7%°C$^{-1}$. The probability density of $\alpha$-values over the tropics is depicted in Supplementary Fig. 2.

**Exemplary temperature histories over tropical oceans**. An exemplary temperature history of a set of rainfall episodes (defined as consecutive 3-hourly rainfall events without interruption, see Methods) precipitating in JASO over a box in the Caribbean Sea (cyan box in Fig. 1a) is depicted in Fig. 2a. The temperatures and rolling 24 h mean temperatures show a very similar behaviour: for all intensity groups (i.e. rainfall episodes within a certain percentile range of rainfall intensities), the temperature and rolling 24 h mean temperature at the beginning of the history (at t = −48 and t = −24 h, respectively) is nearly the same with ~27.8°C. Towards the onset of the rainfall episodes, temperatures and rolling 24 h mean temperatures drift further and further apart, and we observe a stronger decline of the preceding temperatures with increasing intensities of the subsequent rainfall episodes. For the most intense rainfall episodes, the temperature (rolling 24 h mean temperature) falls nearly 0.90 °C (0.35 °C) over the course of the 48 (24) hour history. As a result, the most (least) intense rainfall occurs at the lowest (highest) temperatures at the time of the onset. This is also reflected in the scaling between temperature and extreme-rainfall intensity (Fig. 2c). Here, we place the rainfall episodes pooled from the box in the Caribbean Sea into bins according to their rolling 24 h mean temperature at t = 2 hours before their onset ($T^r$). For each bin, the 90th percentile of rainfall rates ($P^{90}$) is computed, and an exponential regression is applied. A decline in the intensity by 33.7% ± 1.2% (SE) per 1 °C is observed (with a Pearson correlation coefficient (PCC) of −0.981), which is nearly five times the reversed CC rate.

We obtain a quite different result when considering the temperature history of episodes pooled from a box in the northern tropical Atlantic (white box in Fig. 1a), as depicted in Fig. 2b. The temperatures (rolling 24 h mean temperatures) for the different intensity groups show a clear separation at the beginning of the history, which is maintained until around 12 (6) h before the onset of the episodes. From that point onward, temperature curves start to converge until they almost merge at t = 0. The rolling 24 h mean temperature curves exhibit a weaker and delayed convergence, and they maintain their separation at t = 0. This leads to the strongest (weakest) rainfall episodes occurring at the highest (lowest) temperatures, which is consistent with the scaling between the rolling 24 h mean temperatures and extreme-rainfall intensities (Fig. 2d). We find an increase in the intensity of 72.4% ± 1.3% (SE) per 1°C (with a

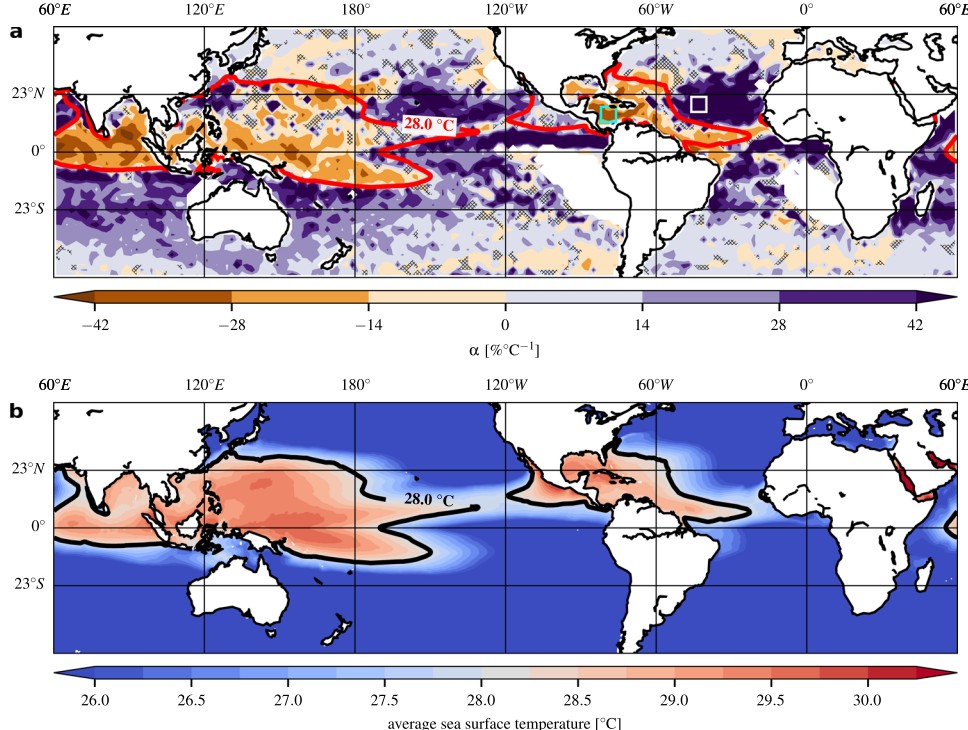

**Fig. 1 Temperature-rainfall scaling and mean sea surface temperatures. a** Spatial pattern of the relationship between temperatures $T^r$ and rainfall intensities $P^{90}$ over water bodies in July–August–September–October (JASO), in terms of fitted $\alpha$-values (in [%°C$^{-1}$]). The colormap ranges from $-6 \times$ CC ($-42$%°C$^{-1}$, brown) to $+6 \times$ CC ($+42$%°C$^{-1}$, purple). Pixels over landmasses, and pixels over water bodies with less than 100 data points per bin, are depicted as white. Pixels with a $p$-value larger than 5% are shaded with crossed black lines. **b** Average sea surface temperatures in JASO over the period from 1998 to 2018. The black line illustrates the 28°C contour line.

PCC of 0.923), more than ten times the rate that would be expected from the thermodynamic CC relationship.

**Correlation between temporal temperature gradients and extreme-rainfall intensity.** Although the temperature histories for the two boxes result in opposed temperature-rainfall scaling behaviours, they have one common feature that stands out: they all show declining temperatures before the onset of the episodes, and the strength of this decline appears to correlate positively with the intensity of the episodes. We emphasize here that this cooling is not caused by prior rainfall, because we have selected only rainfall events for this analysis for which there was no rainfall in the 48 hours before their onset. To further investigate this pre-rainfall cooling, we compute the temporal temperature gradients for all episodes (via the slope of a linear regression through the rolling 24 h mean temperatures from 6 to 2 h before the onset of the episode; see Methods for details), and determine how they scale with rainfall rates. Figure 2e, f depicts the correlation between the temporal temperature gradient and the rainfall intensity for the box in the Caribbean Sea and the northern tropical Atlantic, respectively. They show qualitatively very similar behavior: highest intensities occur for the strongest pre-rainfall temperature decline, weakest intensities for unchanging temperatures. In between, the scaling is linear, with a PCC between temperature gradients and rainfall rates equal to $-0.987$ for both the Caribbean Sea and the northern tropical Atlantic box. For positive gradients, there is no discernible correlation.

To visualize the geographic extent of the validity of this linear correlation, we carried out the correlation analysis between temporal temperature gradients and rainfall intensities for each grid cell. As for the two boxes above, we only consider episodes with negative gradients (see Methods for details; see also Supplementary Fig. 3 for a

geographic map of the proportion of episodes preceded by a negative gradient at each location, Fig. 3a for a map of the average temporal temperature gradient at each location, and Supplementary Fig. 4 for a latitudinal profile of temporal temperature gradients). The geographic map of the slopes of the regression analyses is depicted in Fig. 3b. The slope quantifies the sensitivity of the rainfall rate $P^{90}$ to changes in the temporal temperature gradient $T_g^r$ prior to rainfall onset, in units of mmh$^{-1}$/0.1°Ch$^{-1}$. The median sensitivity of all locations over water (with sufficient data points, see caption of Fig. 3b) is $-2.4$ mmh$^{-1}$/0.1°Ch$^{-1}$, i.e. Fifty percent of ocean locations show an increase in the rainfall rate of more than 2.4 mmh$^{-1}$ per 0.1°Ch$^{-1}$ decrease in the temperature gradient. Compared to the temperature-rainfall scaling (Fig. 1a), we find a substantially more homogeneous distribution of scaling factors (slopes) over tropical oceans. Approximately 94% of all locations over oceans have a negative slope, and 50% of the PCCs of the regressions are smaller than $-0.82$ (see Fig. 3c).

Given the wide-ranging validity of the linear correlation between temporal temperature gradients and rainfall intensities, we also performed the regression analysis for episodes pooled from all northern tropical ocean locations (between 0°N and 23°N). Figure 4a shows the relationship between the temporal temperature gradient and the rainfall intensity for this entire region. We find a strong linear correlation (PCC $= -0.998$) with a sensitivity of $-4.02 \pm 0.0088$ (SE) mmh$^{-1}$/0.1°Ch$^{-1}$.

**Robustness of the correlation between temporal temperature gradients and rainfall intensities.** To further verify the robustness of the linear correlation between temporal temperature gradients and rainfall intensities, we performed the regression analysis over northern tropical oceans using a variety of parameter settings: considering episodes with and without rainfall

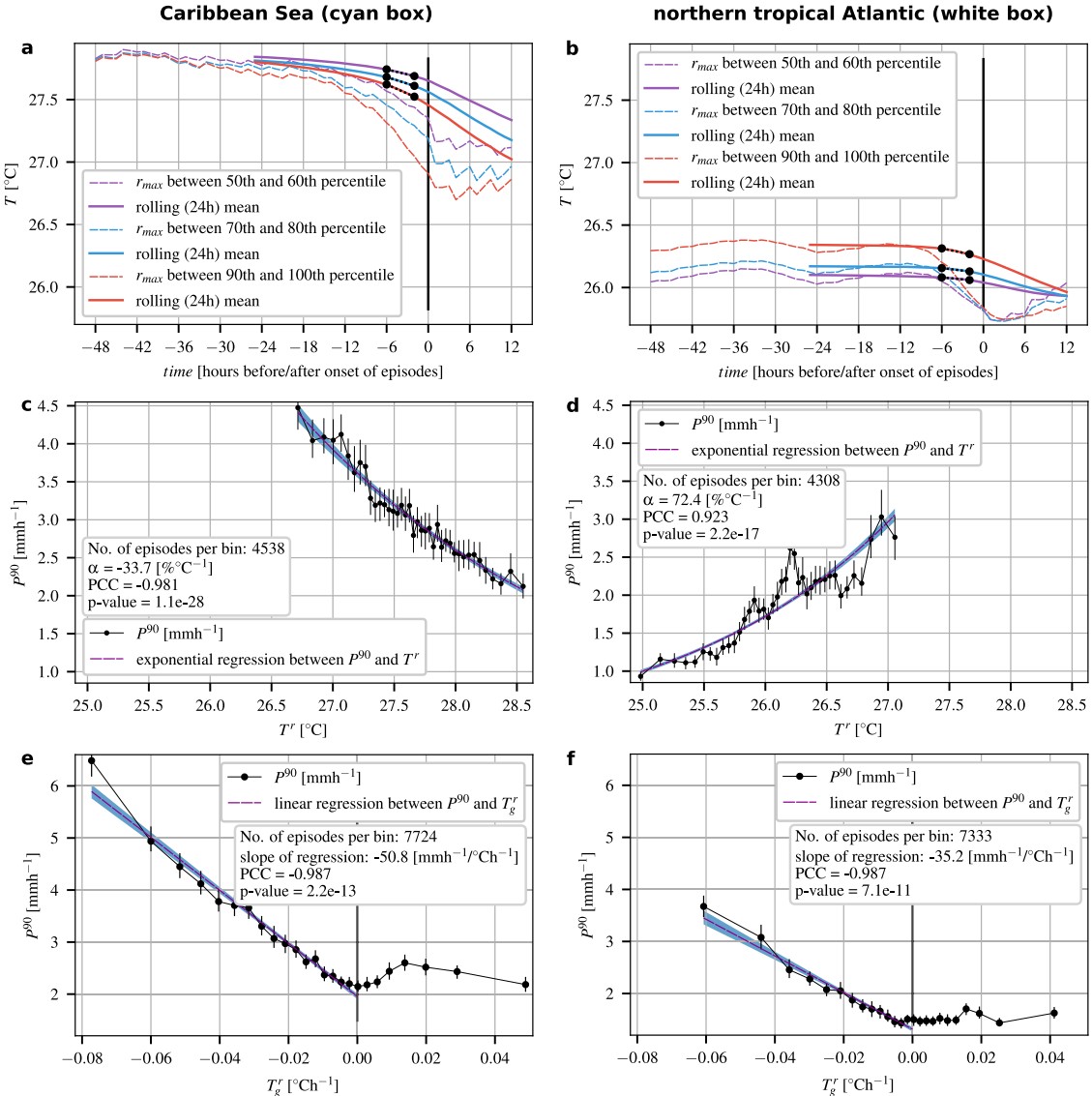

**Fig. 2 Temperature histories and scaling relationships for rainfall episodes over a box in the Caribbean Sea and the northern tropical Atlantic. a** Average temperature history of rainfall episodes over the box in the Caribbean Sea in July–August–September–October (JASO). Episodes are partitioned into groups according to their rainfall intensity $r_{max}$, given by the peak intensity of an episode. All episodes are aligned by the time of their onset. Within each intensity group, the mean temperature is depicted (dashed lines). The rolling 24 h mean of the temperature histories is depicted in solid lines. The dashed black lines represent the regression lines through the rolling 24 h mean temperatures from 6 to 2 h before the onset of the episode. Histories are cut off at 12 h after the onset of the episodes. **b** Same as **a**, but for the box in the northern tropical Atlantic. **c** Observed scaling between temperature ($T^r$) and the 90th percentile of rainfall rates ($P^{90}$) for episodes precipitating in JASO over the box in the Caribbean Sea. Episodes are split into 40 bins according to their temperature ($T^r$), and for each bin, the 90th percentile of rainfall rates is plotted (black solid line and markers; vertical black lines indicate 95% confidence intervals). The exponential regression between $T^r$ and $P^{90}$ is depicted as a dashed magenta line (the blue color indicates the 95% confidence band). **d** Same as **c**, but for the box in the northern tropical Atlantic. **e** Observed scaling between temporal temperature gradients ($T_g^r$) and the 90th percentile of rainfall rates ($P^{90}$) for episodes in JASO sampled from the box in the Caribbean Sea. Episodes are split into 25 bins with respect to their temporal pre-rainfall temperature gradients ($T_g^r$). The 90th percentile for each bin ($P^{90}$) is plotted as a black solid line with markers (vertical black lines indicate 95% confidence intervals). The linear regression through all bins with a negative temporal temperature gradient is depicted as a dashed magenta line (the blue color indicates the 95% confidence band). **f** Same as **e**, but for the box in the northern tropical Atlantic.

before the onset of the episodes (see Supplementary Fig. 5); considering the 95th and 99th percentile of rainfall intensities (see Supplementary Fig. 6); using different temporal window sizes and locations to compute the temperature gradient (see Supplementary Fig. 7); and using the average rather than the maximum rainfall rate of an episodes' constituent rainfall events to define the rainfall rate of an episode (see Supplementary Fig. 8). Although the sensitivity for different parameters varies within

expected ranges, the linearity of the scaling is very robust (the lowest PCC of all parameter configurations is still 0.990). With regards to the general robustness of our results, it should also be noted that the TRMM rainfall product has been shown to underestimate extreme rainfall compared to finer-scale radar-based rainfall datasets[45]. Although this might affect the scaling analyses performed in this study, there is no better alternative to the TRMM product over open oceans at this time.

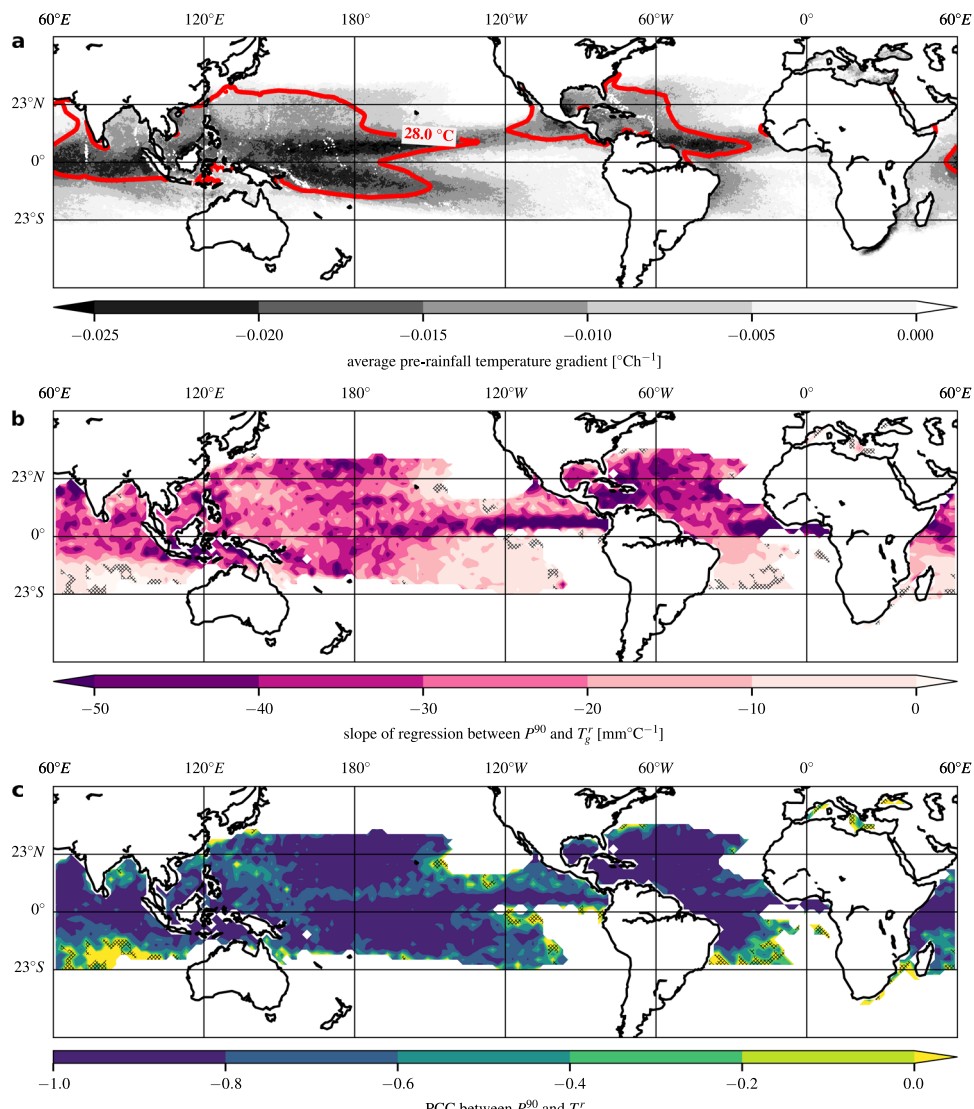

**Fig. 3 Average temporal temperature gradients, scaling factors between temperature gradients and rainfall intensities and their corresponding linear correlation values. a** Average pre-rainfall temperature gradient per pixel ($T_g^r$ in [°Ch$^{-1}$]) for all episodes in July–August–September–October (JASO). **b** Scaling between temporal temperature gradients ($T_g^r$) and extreme-rainfall intensities ($P^{90}$) in JASO. The slope of the linear regression between intensities ($P^{90}$ in [mmh$^{-1}$]) and gradients ($T_g^r$ in [°Ch$^{-1}$]) is indicated by color. The regression is applied considering only those bins for which the representative temperature gradient is negative (see also Fig. 2e, f). Pixels with less than 20 (of a total of 40) such bins, or less than 100 data points per bin, are depicted as white. Pixels over landmasses are also depicted as white. Pixels with a *p*-value larger than 5% are shaded with crossed black lines. **c** Corresponding Pearson Correlation Coefficients (PCCs) of the linear regressions in **b**.

**Results for rainfall episodes associated with tropical cyclones.** We repeated the entire study considering only rainfall episodes that are associated with TCs listed in the International Best Track Archive for Climate Stewardship (IBTrACS) archive (see Methods for details). All tracks in the archive from 1998 to 2018 in JASO are depicted in Supplementary Fig. 9a. Although there are a few studies that investigated the contribution of cyclones to the overall rainfall amount (e.g. refs. [46,47]), those studies mainly focused on contributions over land. Therefore, we computed contributions on a global scale. Supplementary Fig. 9b depicts the proportion of TC-tagged rainfall episodes among all episodes above the 90th percentile of rainfall intensities. Up to 40–50% of episodes are TC-tagged over the northern tropical Atlantic, up to 90–100% over the northeastern Pacific ocean, up to 70–80% over the northwestern Pacific ocean, and up to 10–20% over the northern Indian ocean. The IBTrACS archive only entails a subset of all cyclonic activity, in particular only cyclones with maximum

sustained wind speeds of at least 50–60 kmh$^{-1}$. In view of this fact, the contributions to extreme rainfall overall are surprisingly high.

The results of our analysis using only TC-tagged rainfall episodes are very similar to the results using all episodes. Supplementary Fig. 10 depicts a geographic map of $\alpha$-values over water bodies using only TC-tagged rainfall episodes. Deviations from the thermodynamically expected CC scaling tend to be even larger (in both negative and positive direction) compared to Fig. 1a. Overall, there is more scatter in the spatial variation of $\alpha$-values, which should be expected due to the reduced amount of data. Nevertheless, the geographic patterns of negative and positive temperature-rainfall scaling factors are in good agreement. Temporal pre-rainfall temperature gradients are also predominantly negative over tropical oceans. Finally, we observe nearly the same correlation between temporal temperature gradients and rainfall intensities over northern tropical oceans

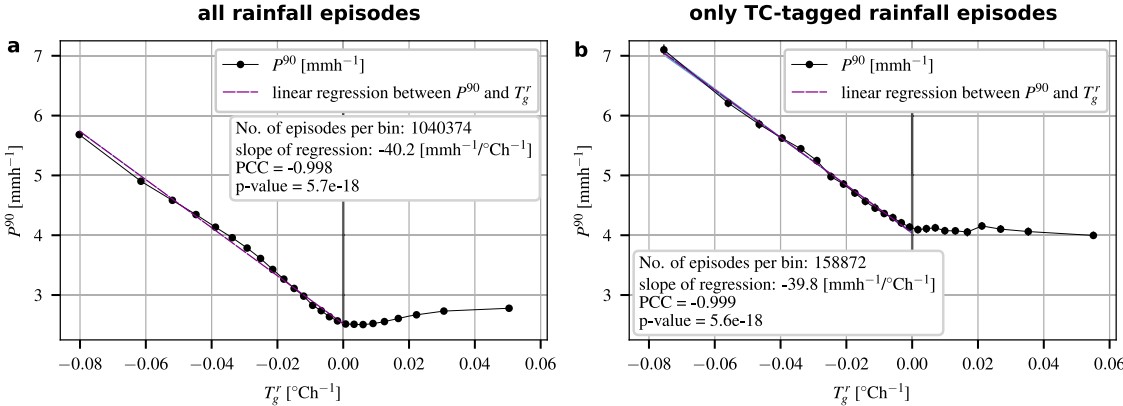

**Fig. 4 Scaling between temporal temperature gradients and extreme-rainfall intensities over northern tropical oceans. a** Observed scaling between temporal temperature gradients ($T_g^r$) and 90th percentiles of rainfall rates ($P^{90}$) for episodes in July–August–September–October (JASO) sampled from all northern tropical ocean locations (between 0°N and 23°N). Rainfall episodes are split into 25 bins with respect to their preceding temporal temperature gradients ($T_g^r$). The 90th percentile for each bin ($P^{90}$) is plotted as a black solid line with markers, the 95% confidence intervals are depicted as vertical black lines, but practically invisible because of their small size. The linear regression through all bins with a negative temporal temperature gradient is depicted as a dashed magenta line. The blue color indicates the 95% confidence interval, but is again too small to be visible. **b** Same as **a**, but restricted to rainfall episodes associated with TCs listed by the IBTrACS archive[57].

for the TC-tagged episodes as for all episodes (compare Fig. 4a with b). Although rainfall rates show higher values overall when we restrict the analysis to TC-tagged episodes, both the Pearson correlation coefficient (PCC = −0.999) and the sensitivity (−3.98 ± 0.0369 (SE) mmh$^{-1}$/0.1°Ch$^{-1}$) remain almost unchanged. Essentially, the only difference between the scaling we obtain using all episodes and the scaling obtained for TC-tagged episodes is the offset on the y-axis.

## Discussion

Although the (nearly) global median of α-values (6.0%°C$^{-1}$ in JASO, and 6.3%°C$^{-1}$ in DJFMA) is close to the expected CC rate of ~7%, we observe a pronounced spatial and seasonal heterogeneity. Particularly over tropical oceans - for which few studies have been published so far (ref. [48] is an exception) - the seasonal variability is apparent (compare Fig. 1a with Supplementary Fig. 11b) and deviations from the CC rate are particularly large, both in positive and negative direction. This indicates that the circulation-dynamical contribution to the temperature-rainfall scaling is several times stronger than the thermodynamic contribution given by the CC relation alone. We emphasize that this is the case even though we took adequate measures to negate the effects potentially influencing the scaling in unintended ways, such as surface cooling by rainfall, seasonality, and the diurnal cycle.

We identify a dynamical mechanism that contributes substantially to the temperature-rainfall scaling over tropical oceans. Most rainfall episodes over tropical oceans show a temperature decline before their onset (Supplementary Fig. 3). The magnitude of this decline is linearly correlated with the subsequent extreme-rainfall intensity (Fig. 4), i.e. the stronger the pre-rainfall temperature declines, the higher the rainfall intensity. This correlation leads to a negative contribution to the temperature-rainfall scaling. In contrast, the CC relation contributes positively to the temperature-rainfall scaling. These two aspects are therefore in competition with each other, contributing in opposing direction to the empirical scaling factors α that we estimate from the data (Fig. 1a). It should also be mentioned that there are without doubt additional (thermo)dynamic contributions to the scaling, given the very strong positive scaling factors over parts of the oceans.

The influence of the intensity-dependent pre-rainfall cooling effect on the temperature-rainfall scaling grows with the proportion of episodes preceded by negative temporal temperature gradients. The strongest pre-rainfall temperature declines are observed in those regions that show a negative temperature-rainfall scaling (compare Fig. 1a and Fig. 3a). This indicates that the contribution of the pre-rainfall cooling effect dominates the scaling behaviour in those regions, leading to a net negative temperature-rainfall scaling. Interestingly, there is also an extensive overlap of these regions with those parts of the tropical oceans that show the highest sea surface temperatures (SSTs) during the study period (see the contour line indicating regions with SSTs above 28 °C in Fig. 1a, b; see also Supplementary Fig. 12 depicting the spatial correlation between long-term average SSTs and α-values).

But why are most rainfall episodes over tropical oceans preceded by negative temperature gradients, and why are those pre-rainfall temperature gradients correlated with the subsequent rainfall intensity? And what is the reason for the extensive spatial overlap between regions with negative scaling and high long-term average SSTs? In order to answer these questions, we need to consider the atmospheric and oceanic dynamics related to tropical cyclones.

First of all, it is well known that the passage of a TC over the ocean causes the upper layers of the ocean to cool substantially[49–51]. The primary factor of this cooling is a mixing of cold water from sub-surface ocean levels with warm surface waters, caused by wind-driven surface divergence.

Additionally, TCs draw in air advectively from surrounding areas, which can extend well into the extratropics[52] where air temperatures are significantly lower than in the proximity of the TC. Cloud cover accompanied by TCs may also play a role, by shielding the ocean surface from direct sunlight before and shortly after the passage of the TC[53]. Temperatures start to decrease up to 2–3 days before the arrival of the cyclone[50,51,54] and the spatial extent of this pre-TC-arrival cooling can reach more than 1.000 km[51]. This explains the pre-rainfall temperature decreases we observe over tropical oceans (Figs. 2a, b and 3a).

Both the amplitude and the spatial extent of the cyclone-related cooling strongly depend on the intensity of the cyclone (defined in terms of its maximum sustained wind speed). The higher the intensity of the cyclone, the larger its spatial extent (see Fig. 6 in[51]) and the stronger the cooling (see Fig. 7a in ref. [51]). Additionally, we find that the extreme-rainfall rate is strongly

correlated with the intensity of cyclones as well (see Supplementary Fig. 13). This is well in line with our finding that the strength of the pre-rainfall temperature decline positively correlates with the subsequent extreme-rainfall intensity. To our knowledge, this relationship has not been quantified before.

Furthermore, the fact that the scaling between temporal temperature gradients and rainfall intensities for all rainfall episodes is essentially a shifted version of the scaling for TC-associated rainfall episodes indicates that the predominant mechanism generating rainfall over oceans is cyclonic activity. There are, of course, other mechanisms that generate rainfall over tropical oceans. For instance, convective rainfall systems near the equator that are non-rotating because of the lack of a sufficiently strong Coriolis force, produce strong rainfall. Two possible reasons that these mechanisms do not noticeably alter the shape of the scaling in Fig. 4a compared to Fig. 4b are: they are statistically outweighed by cyclonic activity; or they are associated with positive temporal pre-rainfall temperature gradients. Further analysis, however, would be required to better understand their influence on the correlation between temperature gradients and rainfall intensities.

The cyclone-related cooling also explains the strong spatial overlap between high SST values and negative temperature-rainfall scaling factors (see Fig. 1a): high SSTs provide a beneficial setting for cyclogenesis. This is reflected in Supplementary Fig. 12, which shows that from 26 °C upwards (approximately the temperature threshold for cyclogenesis[55]), we obtain a negative correlation between long-term average SSTs and $\alpha$-values. Negative $\alpha$-values are heavily concentrated above approximately 28 °C, corresponding to the contour line in Fig. 1a. This is also in line with the observation that the strongest temperature declines occur in those regions with the highest SSTs (see Fig. 3a).

One might expect a larger overlap of these regions with the tropical cyclone tracks provided by IBTrACS (Supplementary Fig. 9a), but a few factors have to be considered in that regard. First, TC tracks only show the propagation of the eye of TCs. The cooling effect, however, may extend up to hundreds of kilometers away from the eye. For instance, even though there are no tracks south of the equator in the TC season of the northern hemisphere, the cooling may very well affect the southern hemisphere as seen in Fig. 3a. Second, not all cyclonic activity is captured in the IBTrACS archive, which only contains tracks of TCs with maximum sustained wind speeds of at least 50–60 kmh$^{-1}$.

In conclusion, our study finds that although the globally averaged temperature-rainfall scaling is indeed close to the thermodynamically expected 7% rainfall intensity increase per 1 °C, there is a pronounced spatial heterogeneity. Hence, on average, the intensity of extreme-rainfall events will increase with rising atmospheric temperatures, which is consistent with the large number of studies on the effect of global warming on rainfall extremes. However, this thermodynamic effect is accompanied by pronounced dynamical effects that lead to complex spatial patterns. To a large extent, the spatial heterogeneity over tropical oceans can be explained by the oceanic and atmospheric dynamics related to cyclonic activity. Our study thus adds to the growing body of research arguing that dynamical effects can strongly influence statistical analyses of temperature-rainfall relationships in observational and simulated data. We believe that dynamical contributions have to be taken into account more thoroughly when investigating temperature-rainfall scaling relationships, in particular in the context of future projections. Especially regarding extreme-rainfall events, we expect model simulations of future projections to exhibit pronounced spatial heterogeneity, as well as substantial deviations from the thermodynamic expectations.

## Methods

**Data**. We use the Tropical Rainfall Measuring Mission (TRMM) 3B42 V7 dataset[43]. It is gridded at a resolution of 0.25° × 0.25° ranging from 50°S to 50°N, and has a 3 hourly temporal resolution. We employ the data for the time period from 1998 to 2018. In order to assure that only data points with significant rainfall are considered in this study, we employ a wet-times threshold of $r \geq 0.1$ mmh$^{-1}$.

In addition, we make use of the temperature of air at two meters above the surface from the ERA5 reanalysis dataset[44]. It has the same spatial resolution (0.25° × 0.25°) as the TRMM rainfall data, with an hourly temporal resolution. We combine this data with the rainfall data as explained in the next section.

For sea surface temperatures (SSTs) from 1998 to 2018 (see Fig. 1b and Supplementary Fig. 12), we employ the NOAA OI SST V2 High Resolution Dataset[56]. It is constructed by combining observations from different platforms (satellites, ships and buoys), and has the same spatial gridding as the TRMM rainfall data (0.25° × 0.25°).

Furthermore, we employ the International Best Track Archive for Climate Stewardship (IBTrACS)[57] to visualize tropical cyclone tracks for the time period from 1998 to 2018 (see Supplementary Fig. 9a), as well as to tag rainfall events as being part of a TC as described in the next section.

**Selection of rainfall episodes to avoid biases in the temperature-rainfall scaling**. The local cooling effect of rainfall on surface temperatures has been shown to influence the scaling behaviour between air temperature and rainfall rates[38]. We therefore take two measures to circumvent any potential influence of this process on our results.

First, for each geographic location of the TRMM grid, we partition the set of rainfall events (above the wet-times threshold of $r \geq 0.1$ mmh$^{-1}$) into episodes, defined as consecutive 3-hourly rainfall events without interruption. We define the rainfall intensity of an episode as the maximum intensity of all rainfall events the episode is comprised of.

Secondly, we only consider episodes without preceding rainfall for at least 48 hours before the onset of the episode.

Additionally, since the annual cycle involves changes in weather and large-scale circulation patterns and therefore rainfall mechanisms in many regions of the world[58], we consider episodes occurring in the TC season of the northern hemisphere (JASO) and the southern hemisphere (DJFMA) separately. Episodes occurring in any other month are discarded in this study. The total number of episodes without preceding rainfall for each geographic location and over the entire study period (in JASO) is depicted in Supplementary Fig. 14.

**Regression analyses between extreme-rainfall intensities, temperature and temporal temperature gradients**. With each rainfall episode, we associate a temperature ($T$) history going back 48 hours from the onset of the rainfall episode. Exemplary histories of rainfall episodes precipitating over the Caribbean Sea and the northern tropical Atlantic in JASO are shown in Fig. 2a, b, respectively.

Based on these histories, we compute two features for each rainfall episode, which will both be used as the independent variables in our scaling analyses. For the conventional temperature-rainfall scaling, we use the 24 h mean temperature two hours before the event, given by $T^r = \frac{1}{24} \cdot \sum_{t=2}^{25} T_t$, where $T_t$ is the temperature $t$ h before the onset of the episode. For our scaling analysis of temporal temperature gradients and rainfall intensities, we use the slope of a linear regression through the rolling 24 h mean temperatures from 6 to 2 h before the onset of the episode (i.e. for the time interval $[-6\,\mathrm{h}, -2\,\mathrm{h}]$), denoted $T_g^r$, with the unit °Ch$^{-1}$. We apply a 24 h rolling mean for a similar reason to why we separate rainfall by seasons, namely to avoid biases induced by the diurnal cycle and specifically to negate the effect of sampling events generated by potentially different rainfall mechanism on the apparent scaling behaviour.

For the scaling analyses, we first bin the temperature (temporal temperature gradient) values into 40 (25) bins with an equal number of samples in each bin. This approach is preferable over using bins of equal width, as it ensures a reliable number of data points across all bins, and avoids sample-size based biases to a great extent[59]. The mean temperature $T^r$ (temporal temperature gradient $T_g^r$) of the events in each bin is used as the representative temperature (gradient) for that bin. We then estimate the 90th percentile of rainfall rates for each bin ($P^{90}$).

With regard to the temperature-rainfall scaling, motivated by the exponential CC relationship, we apply an exponential regression to the rainfall intensities $P^{90}$, by fitting a least-squared linear regression to the logarithm of rainfall intensities. The change in $P^{90}$ with respect to the change in $T^r$ is quantified using the regression between $T^r$ of the first bin and the peak point temperature (the temperature $T^r$ of the bin where the maximum of $P^{90}$ occurred). This relation can be written as: $P_2^{90} = P_1^{90}(1+\alpha)^{(T_2^r - T_1^r)}$, such that $\alpha = 0.068$ is equivalent to a CC like scaling of 6.8%°C$^{-1}$ at 25 °C. Figure 2c, d depict the scaling between $T^r$ and $P^{90}$ for two distinct sets of rainfall episodes. One set of episodes is pooled from the Caribbean Sea (in JASO, Fig. 2c; the box extends from 82°W to 76°W and 12°N to 18°N) and the other set from the northern tropical Atlantic (also in JASO, Fig. 2d; the box extends from 46°W to 40°W and from 16°N to 22°N). Geographic maps of the fitted $\alpha$-values (for JASO) over water bodies and landmasses are shown in Fig. 1a and Supplementary Fig. 1, respectively.

For the scaling between the temporal temperature gradients and rainfall intensities, we apply a least-squared linear regression to all rainfall intensities $P^{90}$, for which the gradient $T_g^r$ is negative. The scaling between $T_g^r$ and $P^{90}$ for the set of rainfall episodes pooled from the Caribbean Sea is illustrated in Fig. 2e, for the set pooled from the northern tropical Atlantic in Fig. 2f, and for the set pooled from all northern tropical oceans in Fig. 4a, b. A geographic map of the slopes of the regression analyses is depicted in Fig. 3b. The corresponding PCCs are shown in Fig. 3c.

In Fig. 2c, d, e, f and Fig. 4a, b, we compute the 95% confidence intervals for the estimates of the 90th percentile of rainfall rates, $P^{90}$, using a bootstrapping approach. We resample the original data 1.000 times (with replacement), and then calculate the 2.5th and 97.5th percentile of the test statistic ($P^{90}$). In the same figures, the regression parameters, i.e. slope and intercept, and their respective standard errors, are computed using quantile regression models[60,61], in particular the Python package Statsmodels[62] and its QuantReg class (statsmodels.regression.quantile_regression.QuantReg). For the 95% confidence bands of the quantile regression lines (same figures), we use a boostrapping approach again (1.000 resamples with replacement).

All p-values stated within figures and figure captions are based on a hypothesis test whose null hypothesis is that the slope of the regression line is zero, using a Wald test with t-distribution of the test statistic. The alternative hypothesis is that the slope of the regression line is nonzero, i.e. we are using a two-sided test.

**Associating rainfall episodes with tropical cyclones**. We essentially conduct our analysis twice: once with all rainfall episodes derived from the TRMM dataset, and another time considering only rainfall episodes associated with TCs listed in the IBTrACS archive. Using the IBTrACS dataset, we tag rainfall episodes as part of a TC whenever they are closer than 1.000 km from the eye of the TC. The proportion of TC-tagged rainfall episodes among all episodes above the 90th percentile of intensity values is depicted in Supplementary Fig. 9b.

**Reporting summary**. Further information on research design is available in the Nature Research Reporting Summary linked to this article.

## Data availability

All data used in this study are publicly available. For rainfall estimates, we used the Tropical Rainfall Measuring Mission (TRMM) 3B42 V7 dataset, available trough https://disc.gsfc.nasa.gov/datasets/TRMM_3B42_7/summary and downloaded from https://disc2.gesdisc.eosdis.nasa.gov/s4pa/TRMM_L3/TRMM_3B42.7/. For temperature estimates, we used the ERA5 reanalysis dataset, downloaded from https://cds.climate.copernicus.eu/cdsapp#!/dataset/reanalysis-era5-single-levels. For sea surface temperatures, we used the NOAA OI SST V2 High Resolution Dataset, available through https://psl.noaa.gov/data/gridded/data.noaa.oisst.v2.highres.html and downloaded from http://ftp.cdc.noaa.gov/Datasets/noaa.oisst.v2.highres/. For tropical cyclone tracks, we used the International Best Track Archive for Climate Stewardship (IBTRACS), available through https://www.ncdc.noaa.gov/ibtracs/ and downloaded from https://www.ncei.noaa.gov/data/international-best-track-archive-for-climate-stewardship-ibtracs/v04r00/access/netcdf/IBTrACS.ALL.v04r00.nc. Source data are provided with this paper.

## Code availability

The Python code used to produce the results and figures of this study is available via GitHub: https://zenodo.org/record/5595864[63].

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

## Acknowledgements

N.B. acknowledges funding by the Volkswagen Foundation, the ClimXtreme project of the BMBF (German Federal Ministry of Education and Research) under grant 01LP1902J and the European Union's Horizon 2020 research and innovation programme under grant agreement No. 820970. The State of Brandenburg (Germany) through the Ministry of Science and Education supported D.T. for part of this study (grant to B.B.).

## Author contributions

D.T. conceived and designed the study with contributions from N.B., A.R. and B.B. D.T. conducted the analysis. D.T. wrote the manuscript with contributions from N.B. All authors discussed and interpreted the results, and edited the manuscript.

## Funding

## Competing interests

The authors declare no competing interests.
