## [Peer Review File · Nature Communications]

The role of cyclonic activity in tropical temperature-rainfall scalingREVIEWER COMMENTS

Reviewer #2 (Remarks to the Author):

The authors attempted to relate rainfall intensity with the temperature to deepen the understanding of the linear dependencies between pre-rainfall cooling and the corresponding cyclone strength, and explain them by atmospheric and oceanic dynamics. The arguments whether the increasing rate of rainfalls particularly extremes follows the CC scaling have attracted many scientific interests. In this regard, the aim of this study is appropriately defined. The objective is well defined with enough background information, and the results are also interesting. However, the methods and analysis show some weaknesses. The manuscript, especially Tropical Cyclone (TC) related analysis lacks scientific importance of this study. So I am skeptical about the publication of this manuscript in the present form.

Major Comments:

I am wondering how this study relates TCs influence on CC-scaling without separating the TC-related bursts? Moreover, it is very much unclear and not at all touched how many bursts are associated with TCs? Authors have also not performed any additional CC-scaling analysis with the (Rainfall, Temperature) pairing samples during the TCs.

L 238-241: I can't understand the logic. Why did authors consider 24 hours rolling and then take the linear trend from -6 to -2 hours? Did they find any significant difference between the two assumptions: 24-hr mean vs. their rolling's linear trend (L247-249)?

L 244-247: In general, the temperature and precipitation follows different probability functions. So it is not convincing to distribute all the samples into an equal number of temperature bins on a global scale. How do authors explain the relatively low/large number of rainfall samples in the tail sides of the corresponding temperature? So I am not confident in the equal number of sampling bins due to the geographical effects (lower temperature over the high latitudes and higher temperature over the equator). This is the reason, the rate of rainfall against temperature quantified in this study contains the contrast of rainfall intensity between the two regions (Fig. 2).

Minor comments:

L 224: Please highlight the threshold (0.1 mm/h) somewhere in this sentence.

TC related discussions in this manuscript are presented to support the results only. So the title of this manuscript seems not appropriate for this study.

Reviewer #3 (Remarks to the Author):

The paper "The Role of Tropical Cyclones in the Temperature-Precipitation Scaling", by Dominik Traxl, Niklas Boers, Aljoscha Rheinwalt and Bodo Bookhagen, addresses the applicability of the Clausius-Clapeyron (CC) relation to infer scaling relations between surface air temperature and extremes of precipitation at local level. The authors find large deviations from the CC scaling in observational data over the tropical oceans, including negative values for the scaling coefficient. They interpret the negative scaling in terms of known processes related to the atmospheric dynamics and air-sea interactions that characterise tropical cyclones. They find (more) robust scalings with indicators still based on surface temperature, but that act as proxies of the intensity of the dynamics relevant for the formation of precipitation rather than of thermodynamic properties of the atmosphere. They thus conclude that purely thermodynamically justified scalings for the response of rainfall extremes to climate change are too simplistic, and that dynamical contributions should also/instead be taken into account.

There is already a relatively large body of literature that identifies deviations from the CC scaling in various regions of the world. This is acknowledged by the authors. The novelty of their results resides in the fact that they identify an impressive amount of spatial heterogeneity and deviation

from the “expected” scaling, in observational data, and over areas relatively less covered by previous studies like the tropical oceans. Additionally, the authors not only explicitly take the position that thermodynamic considerations alone can not explain the observed behaviour, but also identify the relevant dynamical contributions that can instead explain it, and show how these contributions can be to some extent quantified. This work presents therefore a clear and hardly objectionable example of the trappings of adopting simplified thermodynamic arguments for the response of local extremes to climate change, and delivers an important message to the climate community and the broader scientific community. The paper thus fulfils in my opinion the requirements of novelty and relevance for publication on Nature Communications.

The results of the authors are presented in a convincing way and seem quite robust. I have some remarks on the statistical analysis (see below) that I would like the authors to address, but they are mostly a complement and do not question the robustness of the main findings of the paper. I also have some remarks on the presentation of the results, that seems slightly out of focus in parts of the text, as well as some general minor comments on the text (again see below). I am happy to see the paper again after the authors have addressed my comments, if the Editor finds it necessary.

I think that the paper will be an impactful contribution to the conversation on the response of extreme events to climate change. Note that this does not mean claiming the irrelevance of thermodynamic arguments, but rather it will be an input to interpret and investigate them more thoroughly. For example, properly speaking the CC scaling should be sought between precipitation and the condensation temperature of the atmospheric column at which precipitation is formed. Surface temperature is instead used because it is easier to measure and one assumes that it is a good proxy of the relevant atmospheric temperature. A counterargument could therefore be that these results simply state that the near surface temperature in this case is not a good proxy of the atmospheric column temperature, because of a sudden (subdaily scale) temperature drop due to the action of the cyclone on the upper ocean just before the rainfall event. But that if you are able to remove this transient effect somehow, you could recover the CC scaling. In response one could argue that the intensity of the cyclone is instead the dominant factor per se, and that the CC scaling is intrinsically of limited use in this cases. The paper will certainly stimulate discussions and further investigations on this line, and provide a trace of the type of quantitative analysis one can carry on to elucidate these aspects.

Statistical analysis

- 1) can you expand on the definition of your events and how you collect the temperature-rainfall pairs? It is not very clear to me. When you have your rainfall “burst”, say for n consecutive 3 hours periods, and you consider the daily temperature over a 24 hours window ending 2 hours before the onset of the “burst”, which value of precipitation among the n included in the “burst” you associate to that temperature? The first? The maximum? Please explain better this part;
- 2) your definition of an extreme rainfall event with the 90th percentile is quite generous. It would be relevant to see the same analysis performed on the 95th and 99th percentile;
- 3) you say in the caption of the figures that you bin temperature in 40 bins and temperature time derivative in 25 bins, with equal number of samples in each bin, and remove from the analysis grid points where there are less than 20 samples per bin. How many samples do you have per bin typically in the grid points that you analyse? This may reply to my previous question, since if you do not have enough samples you cannot compute higher percentiles. But in that case this should be stated clearly in the text for transparency;
- 4) the shape in figure 2d is reminiscent of the “hook shape” found in similar analysis for example in the Mediterranean region (e.g. Drobinski et al. (2016), Scaling of precipitation extremes with temperature in the French Mediterranean region: What explains the hook shape?, *J. Geophys. Res. Atmos.*, 121, 3100-3119, and references therein). There this non monotone behaviour is explained essentially by seasonality, that cannot be your case (also your data are on the ocean and so they are clearly different). Do you have an interpretation for this behaviour?

5) would it be possible (maybe in the Supplementary Materials) to see a pdf of the values of alpha over the ocean? It is difficult to be sure from the map, but it looks like the distribution may be bimodal. In particular, in the tropical region (say below 23°, where your test cases and most the data you use later are) it seems that the range 0-14 %C⁻¹, where the CC value is, is actually the least populated, thus further supporting your results;

6) although the linear scaling with the temperature time derivative is impressive and seems quite robust, I am slightly worried by the fact that the way you compute the slope is quite arbitrary. In particular, you have a time series filtered over a running window (24 hours), and you compute a linear regression on a range (4 hours) that is much smaller than the filter size. If you change the size of the filter the slope changes too, in particular if you increase it the slope becomes necessarily flatter. How robust is the linearity of the temperature derivative-precipitation scaling (even if with different coefficients) with respect to changes in the size of the filter, and in the size and location of the range where you compute the linear regressions? Can you provide or report on a limited sensitivity analysis on this, for sake of precision?

Presentation and comments on the text (by line index)

51 - "extreme rainfall intensity (i.e., the intensity of rainfall events above the 90th percentile)." I would change the wording: the 90th percentile is not equivalent to extreme in absolute sense or in some standard practice, it's your specific choice;^[1]_{SEP}

60 - define alpha-values. You don't need to introduce a formula, you can refer to the Materials and Methods section for that, but you have at least to say what alpha means (like, "the rate of scaling of extreme rainfall intensity with surface temperature, that would be 6.8%C⁻¹ for a perfect match with the CC relation");

67 - I am a bit perplexed by the fact that you use the term "burst" to define consecutive occurrences of rainfall events. "Burst" refers to something impulsive and transient, which is at odds with the requirement of consecutive, uninterrupted 3 hours periods of rainfall. I understand if you want to use it to identify the transition from a dry time step to a wet time step (that sets your 0 in Fig 2.a-b), but the way you define it here is a bit puzzling to me;

68-69 - the two boxes are very hard to spot on the map, consider a different choice of the color schemes. Also, to guarantee full reproducibility of your results, please include either here or the Supplementary materials the full coordinates of the boxes;

70 - similarly to my comment about the alpha values, you should define "intensity groups", in a short, simple way, leaving technicalities for other sections, but you have to define it or the sentence is impossible to understand. Something like "different percentile ranges of the intensity" could be sufficient;

77 - I wonder whether it is appropriate to use the term "CC scaling plot" for a relation that is quantitatively very far from the proper CC behaviour. I am not aware if this terminology is standard in the literature that focuses on this type of analysis - that is, if any relation with the same functional form as the CC scaling is called a "CC scaling plot" even if the value of alpha is completely different. If not, I think it would be better to call it just "scaling plot" (check the rest of the paper as well);

96 - here it would be useful to introduce a short bridge sentence to explain why you switch to study the time derivative of the temperature drops

96 - please specify that you mean gradients in time. Maybe better, switch to something like "rate of decrease". For a field that depends on space and time, the term gradient is usually associated to the spatial derivatives. Although mathematically it is not incorrect, personally it took me two full reads of the paper to understand that you meant the time derivative;

98 - please avoid using the possessive 's. From here on it is employed several times, please revise the rest of the document;

108 - can you give a simple explanation why this is not necessary to make your point? (I agree that it is not, but "beyond the scope of this study" is very generic);

149 - it is a known process, but references are still needed, e.g. Mei and Pasquero (2013). Spatial and Temporal Characterization of Sea Surface Temperature Response to Tropical Cyclones, Journal of Climate, 26(11), 3745-3765. and references therein (or other papers you could prefer);

175 - "and that the CC scaling behaviour is mostly a reflection of that". Can you better explain what do you mean here?

176-177 - this is incorrect. The scaling is between the temperature derivative and the 90th percentile of the precipitation, so what you are able to predict is the expected value of the rainfall rate of an extreme event if it occurs, not the rainfall rate in general. You would be able to do that if you had done a scaling of the average rainfall rate, not of its 90th percentile. Also, your analysis lacks an estimate of the statistical error you have on the estimate of the 90th percentile, without which it is not very meaningful to talk about prediction.

Reviewer #4 (Remarks to the Author):

Review of NCOMMS-20-37579: The Role of Tropical Cyclones in the Temperature-Precipitation Scaling

Authors: Dominik Traxl, Niklas Boers, Aljoscha Rheinwalt, Bodo Bookhagen

Summary of Manuscript

This manuscript explores the temperature-extreme rainfall relationship and the role of tropical cyclones (TCs) in this scaling. To do this, the authors analyzed a satellite rainfall dataset (TRMM) and a reanalysis temperature dataset (ERA5) over the period of 1998-2018. They found that, despite a close to 7%/oC temperature-extreme rainfall scaling on global average, the spatial pattern presents a strong heterogeneity. They argued that TCs play an important role in the strong negative scaling seen in the tropical oceans.

Summary of my review

The authors aim to explore the role of TCs in the temperature-extreme rainfall scaling which to my knowledge, seems to be fairly novel. This type of research has important implications for an improved understanding of how extreme rainfall responds to global warming. However, I have a couple of concerns. The result section focuses on the spatial pattern of scaling and how it is related to the cooling before the onset of rainfall events, but barely mention TCs. The TC analyses are very qualitative and not convincing which, to my view, cannot meet the standard of a high-profile journal like Nature Communications. I suggest the authors shift the focus to the TC analyses, as will be discussed below. At the current stage, I recommend major revision.

Major Comments

1 as mentioned above, the linkage between TCs and the temperature-extreme rainfall scaling, which should be the main focus of this manuscript – as reflected in the title, lacks in-depth analysis.

1.1 The authors claimed that there is a 'remarkable overlap' between the pattern of strong negative scaling (Fig. 1/3 b) and TC tracks (Fig. 1c) in the western part of the tropical Atlantic and Pacific oceans. First, the overlap is not remarkable to me. For instance, in the Pacific, TC tracks do not extend to tropics up to 180 degree like the negative scaling. Second, this overlap is only part of the picture. a) Why is the eastern tropical Atlantic with considerable TC activity linked with positive scaling? Similarly, the eastern Pacific with strong TC activity is also associated with strong positive scaling. b) Why is the Indian ocean with much less TC activity associated with strong negative scaling although I think there seem to be more storms? The authors also mentioned ITCZ, but barely discussed how it may compound the effect due to TCs. In summary, based on the

visual check, to me the relationship between the TC activity and negative scaling is complex rather than 'remarkable overlap'.

1.2 Quantitative analyses are needed. Instead of tracks, I would suggest using track density, which can better highlight the TC activity. It is also interesting to check if there is a significant anti-correlation between TC density and scaling. Intense storms tend to produce stronger cold wake, so it is interesting to find ways to incorporate TC intensity into the analysis. With the best track data, the authors can create a dataset including both rainfall and temperature without TC and compare this with the original one to see if you get stronger negative scaling.

1.3 The analyses here focus on pre-rainfall cooling. However, the cold wake of TCs typically occurs after the passage of the storm (e.g., Zhang et al., 2019). How to connect the two is another point concerning me.

2 The definition of bursts. The authors only consider 'bursts without preceding rainfall for at least 48 hours'. Does it mean that only the onset three hours of rainfall in an extreme rainfall events with storm period longer than three hours is used as a sample? If so, many samples with extreme rainfall intensity would be excluded from the analysis. Please clarify.

3 There are barely any references starting from the Result section. It is thus not clear how your results fit in previous work. Are they consistent or inconsistent with previous work? I think such information is a vital component of a manuscript, which can connect your work with the field.

Less major comments

1 TRMM dataset tends to overestimate light rainfall while underestimate extreme rainfall. A study (Villarini et al., 2011) shows that the underestimation of TC rainfall in TRMM can be large compared to finer-scale radar-based rainfall datasets. The uncertainty of the TRMM dataset can have an impact on the scaling. I do not expect the authors to resolve this issue, but some discussion on it would be useful.

2 Line 9. It should be temperature-extreme rainfall scaling.

3 Line 141. the intensities of the bursts determine the strength of the decline. Is the causal relation true?

4 Line 240 and probably other places. '-6 to -2 hours before the onset of the burst' is quite confusing. Should it be -6 to -2 hours from...

5 Figures. The caption of Fig. 2 and afterward should mention that the plots are based on the two boxes in Fig. 1b.

Reference

Zhang, J., Lin, Y., Chavas, D. R., & Mei, W. (2019). Tropical cyclone cold wake size and its applications to power dissipation and ocean heat uptake estimates. *Geophysical Research Letters*, 46, 10177– 10185. <https://doi.org/10.1029/2019GL083783>

Villarini, G., Smith, J. A., Baeck, M. L., Marchok, T., and Vecchi, G. A. (2011), Characterization of rainfall distribution and flooding associated with U.S. landfalling tropical cyclones: Analyses of Hurricanes Frances, Ivan, and Jeanne (2004), *J. Geophys. Res.*, 116, D23116, doi:10.1029/2011JD016175.

Responses to reviewer # 2

Reviewer #2 Remarks to the Author:

The authors attempted to relate rainfall intensity with the temperature to deepen the understanding of the linear dependencies between pre-rainfall cooling and the corresponding cyclone strength, and explain them by atmospheric and oceanic dynamics. The arguments whether the increasing rate of rainfalls particularly extremes follows the CC scaling have attracted many scientific interests. In this regard, the aim of this study is appropriately defined. The objective is well defined with enough background information, and the results are also interesting. However, the methods and analysis show some weaknesses. The manuscript, especially Tropical Cyclone (TC) related analysis lacks scientific importance of this study. So I am skeptical about the publication of this manuscript in the present form.

We thank the reviewer for this accurate assessment of the original version of our manuscript. We agree that there were weaknesses in the analysis, particularly regarding the connection between our results and the specific role of tropical cyclones (TCs), which we have thoroughly addressed in our revised manuscript.

We would like to clarify one potential misunderstanding here already, before going into more detail below: We did not try use the scaling between rainfall intensity and temperature to deepen the understanding of the relationship between pre-rainfall cooling and the corresponding cyclone strength. We rather analyze the scaling between temperature and extreme rainfall (whilst carefully mitigating effects that potentially influence the scaling in unintended ways, e.g. surface cooling, effects related to the seasonal and diurnal cycle), find a strong spatial heterogeneity, and then explain that heterogeneity with the atmospheric and oceanic dynamics related to cyclones.

We thus acknowledge that the paper in its original form was not as clear as it should have been, and by answering the reviewers questions and accordingly revising our manuscript, we hope that our revised paper is a lot clearer and more convincing.

Major Comments:

I am wondering how this study relates TCs influence on CC-scaling without separating the TC-related bursts? Moreover, it is very much unclear and not at all touched how many bursts are associated with TCs? Authors have also not performed any additional CC-scaling analysis with the (Rainfall, Temperature) pairing samples during the TCs.

We thank the reviewer for these very valid points. For the revised manuscript, we have repeated the entire study using only rainfall bursts that we tagged as TC-related using the IBTrACS tropical cyclone archive. The main results and conclusions of our study remain, but the separation of TC-related bursts lead to a substantial improvement of our explanations, interpretations and conclusions.

The results are overall very similar for TC-tagged bursts compared to using all bursts. In the revised manuscript, the geographical map of temperature-rainfall scaling factors using all bursts is depicted in Fig. 1a, and using only TC-tagged bursts in Fig. S11 for comparison. Although the spatial variation of α -values using only TC-tagged bursts is more noisy due to the reduced amount of data, the pattern of negative and positive temperature-rainfall scaling factors is in very good agreement. Furthermore, the correlation between pre-rainfall temperature gradients and extreme rainfall intensities is nearly identical for the two cases (compare Fig. 4a and b). Although rainfall rates show higher values overall using only TC-tagged bursts, the Pearson correlation coefficient is almost the same (PCC=-0.999 for TC-tagged bursts, compared to PCC=-0.998 for all bursts), and so is the slope of the linear regression ($-39.7 \text{ mmh}^{-1}/^{\circ}\text{Ch}^{-1}$ for TC-tagged bursts, compared to $-39.9 \text{ mmh}^{-1}/^{\circ}\text{Ch}^{-1}$ for all bursts). Essentially, the only difference between the scaling using all bursts versus TC-tagged bursts is thus an offset on the y-axis.

In our revised manuscript, the atmospheric and oceanic dynamics accompanied by cyclones and our results are combined much clearer. We now make the following arguments, and support them by appropriate references (see paragraph 4 and following of the Discussion section):

- L299ff.: TCs cause a substantial cooling of the upper layers of the ocean and the atmosphere along their tracks

- L307ff.: The cooling can begin up to 2-3 days ahead of the arrival of a cyclone, and the spatial extent of this pre-arrival cooling can reach more than 1.000 km.
- L312ff.: The amplitude and spatial extent of the cooling correlates strongly with the intensity of a cyclone.
- L316f.: The extreme rainfall rate is also strongly correlated with the intensity of a cyclone (this is a result from our revised study, using TC-tagged rainfall bursts to analyze the correlation between maximum sustained wind speeds of cyclones and the 90th percentile of rainfall rates, see Fig. S14).

We argue that cyclone-related cooling leads to the decline we observe in the temperature histories of rainfall bursts over tropical oceans (see Figs. 2a and b, Fig. 3a and Fig. S4). Furthermore, the correlation between cyclone intensity and cooling as well as rainfall intensities (Fig. S14) explains the scaling that we observe between temperature gradients and rainfall intensities (Fig. 4a and b). L319ff.: “In other words, it is the TC intensity that determines the strength of the pre-rainfall cooling as well as the subsequent extreme rainfall intensity.”

Considering the relationships between TC intensity, TC-related pre-rainfall cooling and subsequent rainfall intensities, we interpret the similarity between the rainfall intensity-temperature gradient scaling for all bursts and TC-tagged bursts (Fig. 4a and b, respectively) as an indication that the predominant mechanism generating rainfall over oceans is cyclonic activity (see paragraph 8 in the Discussion section).

We have also added a figure showing the proportion of TC-tagged rainfall bursts among all bursts above the 90th percentile (see Fig. S10b). Up to 40-50% of bursts are TC-related over the northern tropical Atlantic, up to 90-100% over the northeastern pacific ocean, up to 70-80% over the northwestern pacific ocean, and up to 10-20% over the northern Indian ocean.

When interpreting these numbers, it should be kept in mind that the list from the IBTrACS tropical cyclone archive is most likely not exhaustive. The strongest TCs are likely included there, but our arguments would also be valid for cyclonic activity that does not reach the thresholds to be defined as a TC.

L 238-241: I can't understand the logic. Why did authors consider 24 hours rolling and then take the linear trend from -6 to -2 hours? Did they find any significant difference between the two assumptions: 24-hr mean vs. their rolling's linear trend (L247-249)?

There is a principal difference between using the 24-h mean and the linear trend. Using the 24-h mean, we relate *absolute temperatures* to rainfall intensities, which is common practice in conventional Clausius-Clapeyron (CC) scaling analyses. Relating the trend, or in other words the *temporal temperature gradient* (i.e. the slope of the linear regression) to rainfall intensities has - to our knowledge - not been done before.

We use the 24-h mean to compute the temperature-rainfall scaling factors (α -values) as depicted in Fig. 1a, Fig. S1 and Figs. 2c and d, and find a strong spatial heterogeneity, particularly over tropical oceans. Using temporal temperature gradients, we find a substantially more homogeneous distribution of scaling factors (see Figs. 2e and f and Figs. 3b and c). Over tropical oceans, we discover a robust and perfectly linear correlation between temporal temperature gradients and extreme rainfall intensities (see Figs. 4a and b), which we then explain by the oceanic and atmospheric dynamics accompanied by cyclones (see paragraphs 5, 6 and 7 in the Discussion section of the revised manuscript). We explain the spatial heterogeneity of temperature-rainfall scaling factors with the newly discovered linear relationship between temporal temperature gradients and extreme rainfall intensity (see paragraphs 2 and 3 of the Discussion section of the revised manuscript).

In our revised manuscript, we also put a stronger focus on the separation of temperatures for rainfall bursts in different intensity percentile groups long before the onset of the bursts (at least 24 hours), and argue how this together with the correlation between temporal temperature gradients and rainfall intensities explains the observed α scaling (see also paragraphs 2 and 3 of the Discussion section).

L 244-247: In general, the temperature and precipitation follows different probability functions. So it is not convincing to distribute all the samples into an equal number of temperature bins on a global scale. How do authors explain the relatively low/large number of rainfall samples in the tail sides of the corresponding temperature? So I am not confident in the equal number of sampling bins due to the geographical effects (lower temperature over the high latitudes and higher temperature over the equator). This is the reason, the rate of rainfall against temperature quantified in this study contains the contrast of rainfall intensity

between the two regions (Fig. 2).

It is true that temperature and rainfall intensity have very different probability distributions. We do not, however, distribute all the samples into an equal number of temperature bins on a global scale. Instead, for each geographical location (i.e., locally), we bin the corresponding (Rainfall, Temperature) pairing samples into 40 bins with an *equal number of samples in each bin*. We chose to do it this way precisely to avoid the possible biases mentioned by the referee. In particular, our approach ensures that we do not obtain a different number of samples in the tails of the distributions, and avoids sample-size based biases to a great extent (see L421ff.).

Minor comments:

L 224: Please highlight the threshold (0.1 mm/h) somewhere in this sentence.

We thank the reviewer for the suggestion, and mention the threshold in L58ff., L368f. and L389f. of the revised manuscript.

TC related discussions in this manuscript are presented to support the results only. So the title of this manuscript seems not appropriate for this study.

We agree that in the original manuscript, the TC related analysis was not extensive enough to warrant the title. Considering the substantial TC-related analysis we incorporated in the revised manuscript, we believe the title would be justified now. Nevertheless, we changed the title to “The Role of Cyclonic Activity in the Temperature-Precipitation Scaling”, since we interpret the similarity of the results for TC-tagged bursts and all bursts as an indication that it is the influence of all cyclonic activity (not only tropical cyclones with at least 50-60 km/h maximum sustained wind speeds, as captured by the IBTrACS archive) that is reflected in the heterogeneity of α -values over tropical oceans.

Responses to reviewer # 3

Reviewer #3 Remarks to the Author:

The paper “The Role of Tropical Cyclones in the Temperature-Precipitation Scaling”, by Dominik Traxl, Niklas Boers, Aljoscha Rheinwalt and Bodo Bookhagen, addresses the applicability of the Clausius-Clapeyron (CC) relation to infer scaling relations between surface air temperature and extremes of precipitation at local level. The authors find large deviations from the CC scaling in observational data over the tropical oceans, including negative values for the scaling coefficient. They interpret the negative scaling in terms of known processes related to the atmospheric dynamics and air-sea interactions that characterise tropical cyclones. They find (more) robust scalings with indicators still based on surface temperature, but that act as proxies of the intensity of the dynamics relevant for the formation of precipitation rather than of thermodynamic properties of the atmosphere. They thus conclude that purely thermodynamically justified scalings for the response of rainfall extremes to climate change are too simplistic, and that dynamical contributions should also/instead be taken into account.

There is already a relatively large body of literature that identifies deviations from the CC scaling in various regions of the world. This is acknowledged by the authors. The novelty of their results resides in the fact that they identify an impressive amount of spatial heterogeneity and deviation from the “expected” scaling, in observational data, and over areas relatively less covered by previous studies like the tropical oceans. Additionally, the authors not only explicitly take the position that thermodynamic considerations alone can not explain the observed behaviour, but also identify the relevant dynamical contributions that can instead explain it, and show how these contributions can be to some extent quantified. This work presents therefore a clear and hardly objectionable example of the trappings of adopting simplified thermodynamic arguments for the response of local extremes to climate change, and delivers an important message to the climate community and the broader scientific community. The paper thus fulfils in my opinion the requirements of

novelty and relevance for publication on Nature Communications.

The results of the authors are presented in a convincing way and seem quite robust. I have some remarks on the statistical analysis (see below) that I would like the authors to address, but they are mostly a complement and do not question the robustness of the main findings of the paper. I also have some remarks on the presentation of the results, that seems slightly out of focus in parts of the text, as well as some general minor comments on the text (again see below). I am happy to see the paper again after the authors have addressed my comments, if the Editor finds it necessary.

I think that the paper will be an impactful contribution to the conversation on the response of extreme events to climate change. Note that this does not mean claiming the irrelevance of thermodynamic arguments, but rather it will be an input to interpret and investigate them more thoroughly. For example, properly speaking the CC scaling should be sought between precipitation and the condensation temperature of the atmospheric column at which precipitation is formed. Surface temperature is instead used because it is easier to measure and one assumes that it is a good proxy of the relevant atmospheric temperature. A counterargument could therefore be that these results simply state that the near surface temperature in this case is not a good proxy of the atmospheric column temperature, because of a sudden (subdaily scale) temperature drop due to the action of the cyclone on the upper ocean just before the rainfall event. But that if you are able to remove this transient effect somehow, you could recover the CC scaling. In response one could argue that the intensity of the cyclone is instead the dominant factor per se, and that the CC scaling is intrinsically of limited use in this cases. The paper will certainly stimulate discussions and further investigations on this line, and provide a trace of the type of quantitative analysis one can carry on to elucidate these aspects.

We are glad to read that the referee recommends our manuscript for publication, and want to thank them for the detailed comments, critique and recommendations. Please find our responses to the points raised, as well as references to according changes in the manuscript, below.

Statistical analysis:

1) can you expand on the definition of your events and how you collect the temperature-rainfall pairs? It is not very clear to me. When you have your rainfall "burst", say for n consecutive 3 hours periods, and you consider the daily temperature over a 24 hours window ending 2 hours before the onset of the "burst", which value of precipitation among the n included in the "burst" you associate to that temperature? The first? The maximum? Please explain better this part;

Thank you for this careful observation. In the original manuscript, the definition of the intensity of rainfall bursts was indeed quite hidden in the caption of Fig. 2a. In the revised manuscript, we have added this information in L392f.: "We define the rainfall intensity of a [burst] as the maximum intensity of all rainfall events the [burst] is comprised of."

As part of a "robustness" analysis we added to the revised manuscript, we also investigated the rainfall intensity-temperature gradient scaling using the average rainfall intensity to define the intensity of a burst (Fig. S9). Although the slope of the linear regression between temporal temperature gradients and extreme rainfall intensities is different, the correlation is still perfectly linear with a Pearson correlation coefficient of -0.998.

2) your definition of an extreme rainfall event with the 90th percentile is quite generous. It would be relevant to see the same analysis performed on the 95th and 99th percentile;

We thank the reviewer for the suggestion. As mentioned in the answer to the last question, we have added a "robustness" paragraph to the revised manuscript that also entails an analysis of the rainfall intensity-temperature gradient scaling using the 95th and 99th percentile of rainfall intensities (see paragraph 9 of the Results section, and in particular Figs. S7a and b). Although the slope of the regression changes for different percentiles, the correlation is still almost perfectly linear. The lowest Pearson correlation coefficient is -0.990 for the 99th percentile.

We have also performed the global temperature-rainfall scaling analysis using the 95th and 99th percentile, but have not included it in the manuscript for brevity. Below, you find the geographical map of temperature-

rainfall scaling factors using the 90th, 95th and 99th percentile of rainfall intensities. Additionally, we depicted the number of points per bin available to estimate the percentile values for the temperature-rainfall scaling analysis. Please note that the red line corresponds to the 28°C contour line of average sea surface temperatures, as depicted in Fig. 1b in the revised manuscript.

The spatial distribution of positive and negative α -values is in very good agreement for the different percentile values. Positive scaling factors, however, tend towards larger values with increasing percentiles.

3) you say in the caption of the figures that you bin temperature in 40 bins and temperature time derivative in 25 bins, with equal number of samples in each bin, and remove from the analysis grid points where there are less than 20 samples per bin. How many samples do you have per bin typically in the grid points that you analyse? This may reply to my previous question, since if you do not have enough samples you cannot compute higher percentiles. But in that case this should be stated clearly in the text for transparency;

Please see the lower right plot of the above figure depicting the number of points per bin for the temperature-rainfall scaling analysis. For the rainfall intensity-temperature gradient analysis, we just need to multiply the numbers by a factor of $40/25=1.6$. For the vast majority of locations, the amount of data is sufficient to get a reasonable estimate of higher percentiles (e.g., the 99th percentile shown above for the temperature-rainfall scaling analysis). In the revised manuscript, we have added a figure depicting the total number of bursts considered in the study (Fig. S15).

4) the shape in figure 2d is reminiscent of the “hook shape” found in similar analysis for example in the Mediterranean region (e.g. Drobinski et al. (2016), Scaling of precipitation extremes with temperature in the French Mediterranean region: What explains the hook shape?, *J. Geophys. Res. Atmos.*, 121, 3100-3119, and references therein). There this non monotone behaviour is explained essentially by seasonality, that cannot be your case (also your data are on the ocean and so they are clearly different). Do you have an interpretation for this behaviour?

First, please note that the “hook shape” does not appear in the revised manuscript anymore, since we are now using data from July–August–September–October (JASO, the tropical cyclone season in the northern hemisphere), rather than June–July–August (JJA). Also, we shifted the coordinates of the box in the northern tropical Atlantic slightly. Please see Fig. 2d for the temperature-rainfall scaling plot of that box.

Nevertheless, we did observe this shape in some locations we have analyzed in preparing and working on this study. Generally speaking, we found a few factors that could lead to such a shape:

- seasonality: using rainfall events sampled irrespective of seasons can lead to very different temperature-rainfall scaling factors. For instance, the temperature-rainfall scaling factor for a certain location could

be positive for both JJA and DJF, but negative considering JJA and DJF combined. For this reason, we have separated bursts occurring in different seasons in our study.

- diurnal cycle: we have observed a similar effect when sampling rainfall events irrespective of the diurnal cycle, i.e. the time of the day the rainfall event occurred. For instance, for some locations, we have observed that the highest intensities always occur at a certain time of the day, which is not necessarily the time of the day showing the highest temperature. Depending on which temperature one uses to form temperature-intensity pairs for the temperature-rainfall scaling analysis, this can lead to very different results. That is why we use the 24h rolling mean temperature in our analysis, because it effectively negates this problem.
- lack of available moisture: another factor is the potential lack of available moisture in high temperature regimes. This is particularly true for locations over land. However, we have not investigated this in our study.
- dynamical effects: theoretically, it is of course possible that some dynamical contributions lead to “hook shapes”, just like they can lead to negative scaling over large parts of tropical oceans as observed in our study.

5) would it be possible (maybe in the Supplementary Materials) to see a pdf of the values of alpha over the ocean? It is difficult to be sure from the map, but it looks like the distribution may be bimodal. In particular, in the tropical region (say below 23°, where your test cases and most the data you use later are) it seems that the range 0-14 %C⁻¹, where the CC value is, is actually the least populated, thus further supporting your results;

We thank the referee for this suggestion and have added a figure of the probability density of α -values over tropical oceans accordingly (see Fig. S2). As presumed by the referee, the range 0-14 %C⁻¹ is much less populated than the regions with lower/higher α -values.

6) although the linear scaling with the temperature time derivative is impressive and seems quite robust, I am slightly worried by the fact that the way you compute the slope is quite arbitrary. In particular, you have a time series filtered over a running window (24 hours), and you compute a linear regression on a range (4 hours) that is much smaller than the filter size. If you change the size of the filter the slope changes too, in particular if you increase it the slope becomes necessarily flatter. How robust is the linearity of the temperature derivative-precipitation scaling (even if with different coefficients) with respect to changes in the size of the filter, and in the size and location of the range where you compute the linear regressions? Can you provide or report on a limited sensitivity analysis on this, for sake of precision?

We thank the referee for bringing up their concern about the robustness of the rainfall intensity-temperature gradient scaling. As part of the aforementioned “robustness” analysis we have added to the revised manuscript (see paragraph 9 of the Results section), we have also looked at the influence of the time-window used to compute temperature gradients on the rainfall intensity-temperature gradient scaling (see Fig. S8). We have not changed the filter size of 24h, since its function is to negate the effects of the diurnal cycle on the scaling analyses. We did, however, change the location and size of the time-window we use to compute the gradient. In addition to the time-window [6 to 2] hours before the onset of the burst we use in the main text, we considered the time-windows [12 to 2], [18 to 2] and [24 to 12]. Although the slope of the regression (expectedly) changes, the linearity of the scaling is very robust (the lowest Pearson correlation coefficient of all time-windows used is -0.992).

Presentation and comments on the text (by line index)

51 - “extreme rainfall intensity (i.e., the intensity of rainfall events above the 90th percentile).“ I would change the wording: the 90th percentile is not equivalent to extreme in absolute sense or in some standard practice, it’s your specific choice;

We have changed the wording to “We define extreme rainfall as rainfall events above the 90th percentile of wet times, i.e., 3-hourly rainfall events with average rainfall rates above 0.1 mmh⁻¹.” (L59ff.).

60 - define alpha-values. You don't need to introduce a formula, you can refer to the Materials and Methods section for that, but you have at least to say what alpha means (like, "the rate of scaling of extreme rainfall intensity with surface temperature, that would be $6.8\%C^{-1}$ for a perfect match with the CC relation");

We rephrased the sentence in which α -values first occur to: "We applied an exponential regression between temperature and extreme rainfall intensity for each grid cell of the data covering the globe from $50^{\circ}S$ to $50^{\circ}N$; the resulting scaling factors (α -values, in units of $\%C^{-1}$, see Methods) are depicted in Figs. 1a and S1." (L76ff.).

67 - I am a bit perplexed by the fact that you use the term "burst" to define consecutive occurrences of rainfall events. "Burst" refers to something impulsive and transient, which is at odds with the requirement of consecutive, uninterrupted 3 hours periods of rainfall. I understand if you want to use it to identify the transition from a dry time step to a wet time step (that sets your 0 in Fig 2.a-b), but the way you define it here is a bit puzzling to me;

We thank the referee for sharing their concern regarding the choice of the term "burst". In the revised manuscript, we changed every occurrence of the term "rainfall burst" to "rainfall episode".

68-69 - the two boxes are very hard to spot on the map, consider a different choice of the color schemes. Also, to guarantee full reproducibility of your results, please include either here or the Supplementary materials the full coordinates of the boxes;

We agree that the visibility of the boxes could be improved, and changed the color of the box in the northern tropical Atlantic to white (see Fig. 1a). We also included the coordinates of the two boxes in L437ff.

70 - similarly to my comment about the alpha values, you should define "intensity groups", in a short, simple way, leaving technicalities for other sections, but you have to define it or the sentence is impossible to understand. Something like "different percentile ranges of the intensity" could be sufficient;

We completely agree with the reviewer and added an explanation in L93ff.: "[.] for all intensity groups (i.e., rainfall episodes within a certain percentile range of rainfall intensities) [..]"

77 - I wonder whether it is appropriate to use the term "CC scaling plot" for a relation that is quantitatively very far from the proper CC behaviour. I am not aware if this terminology is standard in the literature that focuses on this type of analysis - that is, if any relation with the same functional form as the CC scaling is called a "CC scaling plot" even if the value of alpha is completely different. If not, I think it would be better to call it just "scaling plot" (check the rest of the paper as well);

Although the terms "CC scaling analysis" and "CC scaling plot" are quite common in the literature (even if the actual scaling turns out to be rather different to the thermodynamically expected $7\%/^{\circ}C$), we agree with the referee that the usage of these terms is misleading. Therefore, we changed all occurrences of "CC scaling" to "temperature-rainfall scaling", apart from when we are referring to the thermodynamic CC relationship.

96 - here it would be useful to introduce a short bridge sentence to explain why you switch to study the time derivative of the temperature drops

We agree with the referee. Over the course of the revision, we have added a few paragraphs to the results section, and rearranged the order. In the revised manuscript, we introduce the rainfall intensity-temperature gradient scaling directly after describing the temperature declines in the temperature history plots of the two boxes (Fig 2a and b), see paragraph 6 of the Results section. Only after that - motivated by the linear correlation observed in the boxes - we describe the global analysis of rainfall intensity-temperature gradient scaling factors.

96 - please specify that you mean gradients in time. Maybe better, switch to something like "rate of decrease". For a field that depends on space and time, the term gradient is usually associated to the spatial

derivatives. Although mathematically it is not incorrect, personally it took me two full reads of the paper to understand that you meant the time derivative;

The first time we introduce the term “gradient” in the revised manuscript, we added a clear description of what is meant by it. L150ff.: “To further investigate this pre-rainfall cooling, we compute the temporal temperature gradients for all episodes (via the slope of a linear regression through the rolling 24-hour mean temperatures from 6 to 2 hours before the onset of the episode; see Methods for details), [..].” Whenever the term “gradient” is mentioned in the revised manuscript, we added the term “temporal” in front of it. We would prefer to stick to the term “gradient”, since it’s occurring very frequently and we could not think of a term that is as short as that.

98 - please avoid using the possessive 's. From here on it is employed several times, please revise the rest of the document;

We thank the referee for this linguistic hint. In the revised manuscript, we do not make use of the possessive 's anymore.

108 - can you give a simple explanation why this is not necessary to make your point? (I agree that it is not, but “beyond the scope of this study” is very generic);

The phrase has been removed in the revised manuscript. Since the focus of this paper is tropical oceans and the extratropics are practically not analyzed in the study at all, we do not think that this sentence adds anything meaningful.

149 - it is a known process, but references are still needed, e.g. Mei and Pasquero (2013). Spatial and Temporal Characterization of Sea Surface Temperature Response to Tropical Cyclones, *Journal of Climate*, 26(11), 3745-3765. and references therein (or other papers you could prefer);

We thank the reviewer for the suggested references. In the revised manuscript, the atmospheric and oceanic dynamics accompanied by cyclones and our results are combined much clearer. We now make the following arguments, and support them by appropriate references (paragraph 4 and following of the Discussion section):

- L299ff.: TCs cause a substantial cooling of the upper layers of the ocean and the atmosphere along their tracks
- L307ff.: The cooling can begin up to 2-3 days ahead of the arrival of a cyclone, and the spatial extent of this pre-arrival cooling can reach more than 1.000 km.
- L312ff.: The amplitude and spatial extent of the cooling correlates strongly with the intensity of a cyclone.
- L316f.: The extreme rainfall rate is also strongly correlated with the intensity of a cyclone (this is a result from our revised study, using TC-tagged rainfall bursts to analyze the correlation between maximum sustained wind speeds of cyclones and the 90th percentile of rainfall rates, see Fig. S14).

We argue that cyclone-related cooling leads to the decline we observe in the temperature histories of rainfall bursts over tropical oceans (see Figs. 2a and b, Fig. 3a and Fig. S4). Furthermore, the correlation between cyclone intensity and cooling as well as rainfall intensities (Fig. S14) explains the scaling that we observe between temperature gradients and rainfall intensities (Fig. 4a and b). L319ff.: “In other words, it is the TC intensity that determines the strength of the pre-rainfall cooling as well as the subsequent extreme rainfall intensity.”

175 - “and that the CC scaling behaviour is mostly a reflection of that”. Can you better explain what do you mean here?

We admit that this sentence in the original manuscript is not exactly self-explanatory, and thank the reviewer for pointing it out. In the revised manuscript, this sentence is removed and replaced by a more

extensive description of how the temperature-rainfall scaling factors are influenced by the rainfall intensity-temperature gradient scaling. Essentially, the spatial distribution of temperature-rainfall scaling factors over tropical oceans is the result of an interplay between two factors (see paragraph 2 and following of the Discussion section).

- The separation of temperatures for rainfall bursts in different intensity percentile groups long before the onset of the bursts (at least 24 hours). This separation is much weaker in regions with pronounced negative temperature-rainfall scaling than in regions with strongly positive scaling (exemplarily shown in Fig. 2a and b, but confirmed for many boxes over tropical oceans). We argue that this is primarily due to a saturation effect of temperatures in those regions with negative temperature-rainfall scaling, which coincide with regions that show the highest average sea surface temperatures (Fig. 1b) as well as low daily temperature variation (Fig. S3).
- The temperature decline before the onset of rainfall bursts, and the correlation of the strength of this decline with the subsequent extreme rainfall intensity. Nearly all tropical ocean locations show predominantly negative pre-rainfall temperature gradients (see Fig. S4). For bursts with a negative temperature gradient, we find a linear correlation between the magnitude of the temperature decline and the extreme rainfall intensity. This correlation leads to a separation (convergence) of the temperature curves of bursts in different intensity groups, as exemplarily shown in Fig. 2a (Fig. 2b).

Together, these two factors govern the temperature-rainfall scaling over tropical oceans. Low initial temperature separation (at $t=-24$ hours) and the negative correlation of rainfall intensity with temperature gradients (related to the strength of the pre-rainfall cooling) leads to negative temperature-rainfall scaling (Fig. 2a). For locations with high initial temperature separation, the rainfall intensity-temperature gradient correlation actually reduces the strongly positive temperature-rainfall scaling, which would be even higher without the effect of this correlation (Fig. 2b).

176-177 - this is incorrect. The scaling is between the temperature derivative and the 90th percentile of the precipitation, so what you are able to predict is the expected value of the rainfall rate of an extreme event if it occurs, not the rainfall rate in general. You would be able to do that if you had done a scaling of the average rainfall rate, not of its 90th percentile. Also, your analysis lacks an estimate of the statistical error you have on the estimate of the 90th percentile, without which it is not very meaningful to talk about prediction.

We completely agree with the referee, and removed this part in the revised manuscript. Although in principle it would be interesting to assess the predictive power of the linear correlation between pre-rainfall temperature gradients and extreme rainfall intensities, we believe it would take up too much space in this study considering that it is not important for the main message of the paper.

Responses to reviewer # 4

Reviewer #4 Remarks to the Author:

Review of NCOMMS-20-37579: The Role of Tropical Cyclones in the Temperature-Precipitation Scaling

Authors: Dominik Traxl, Niklas Boers, Aljoscha Rheinwalt, Bodo Bookhagen

Summary of Manuscript

This manuscript explores the temperature-extreme rainfall relationship and the role of tropical cyclones (TCs) in this scaling. To do this, the authors analyzed a satellite rainfall dataset (TRMM) and a reanalysis temperature dataset (ERA5) over the period of 1998-2018. They found that, despite a close to $7\%/^{\circ}\text{C}$ temperature-extreme rainfall scaling on global average, the spatial pattern presents a strong heterogeneity. They argued that TCs play an important role in the strong negative scaling seen in the tropical oceans.

Summary of my review

The authors aim to explore the role of TCs in the temperature-extreme rainfall scaling which to my knowledge, seems to be fairly novel. This type of research has important implications for an improved understanding of how extreme rainfall responds to global warming. However, I have a couple of concerns. The result section focuses on the spatial pattern of scaling and how it is related to the cooling before the onset of rainfall events, but barely mention TCs. The TC analyses are very qualitative and not convincing which, to my view, cannot meet the standard of a high-profile journal like Nature Communications. I suggest the authors shift the focus to the TC analyses, as will be discussed below. At the current stage, I recommend major revision.

We thank the reviewer for their constructive critique, comments and recommendations. We acknowledge that there were weaknesses in the study, especially concerning the connection between our results and tropical cyclones. For that matter, we repeated the entire study using only rainfall bursts that we tagged as TC-related using the IBTrACS tropical cyclone archive. Although the main results and conclusions of our study remain, we think that the separation of TC-related bursts lead to a substantial improvement of our explanations, interpretations and conclusions. Please find a point-by-point response to the comments below.

Major Comments

1 as mentioned above, the linkage between TCs and the temperature-extreme rainfall scaling, which should be the main focus of this manuscript – as reflected in the title, lacks in-depth analysis.

1.1 The authors claimed that there is a ‘remarkable overlap’ between the pattern of strong negative scaling (Fig. 1/3 b) and TC tracks (Fig. 1c) in the western part of the tropical Atlantic and Pacific oceans. First, the overlap is not remarkable to me. For instance, in the Pacific, TC tracks do not extend to tropics up to 180 degree like the negative scaling. Second, this overlap is only part of the picture. a) Why is the eastern tropical Atlantic with considerable TC activity linked with positive scaling? Similarly, the eastern Pacific with strong TC activity is also associated with strong positive scaling. b) Why is the Indian ocean with much less TC activity associated with strong negative scaling although I think there seem to be more storms? The authors also mentioned ITCZ, but barely discussed how it may compound the effect due to TCs. In summary, based on the visual check, to me the relationship between the TC activity and negative scaling is complex rather than ‘remarkable overlap’.

We agree that the overlap between TC tracks and the pattern of negative scaling factors is indeed more complex than we described it in the original manuscript, and thank the reviewer for these careful observations. In the revised manuscript, we do not write “remarkable overlap” anymore, but instead explain why the overlap is more complex than one might expect. Let us explain this in more detail.

First, we want to mention that in the revised manuscript, we do not consider bursts precipitating in JJA anymore, but rather in July–August–September–October (JASO), since these are the months with the highest TC activity in the northern hemisphere.

In the revised manuscript, we introduce an additional data set: the NOAA OI SST V2 High Resolution Dataset containing sea surface temperatures (SSTs, see paragraph 3 of the Methods section). The average SSTs in JASO (the tropical cyclone season of the northern hemisphere) over the study period from 1998 to 2018 are depicted as a geographical map in Fig. 1b. Considering the spatial pattern of negative scaling factors, we find a strong overlap with those regions of the tropical oceans that show the highest sea surface temperatures (above 28°C, compare Fig. 1a with Fig. 1b). Although the spatial overlap is not perfect (see, for instance, the positive scaling factors in the north-western part of the Pacific Ocean), it is quite extensive (a quantitative analysis of the overlap will follow below). In the revised manuscript, we explain this overlap by the interplay of two factors (see paragraph 2 and following of the Discussion section).

- The temperature separation of rainfall bursts in different intensity percentile groups long before the onset of the bursts (at least 24 hours).
- The temperature decline before the onset of rainfall bursts, and the correlation of the strength of this decline with the subsequent extreme rainfall intensity.

The first factor has not been discussed in much detail in the original manuscript. However, over the course of the revision, we realized that it is an important ingredient to understand the spatial distribution of scaling factors over tropical oceans. The pre-rainfall temperature separation of bursts in different intensity groups is much weaker in regions with pronounced negative temperature-rainfall scaling than in regions with strongly positive scaling (exemplarily shown in Fig. 2a and b, but confirmed for many boxes over tropical oceans). We argue that this is primarily due to a saturation effect of temperatures in those regions with negative temperature-rainfall scaling, which coincide with regions that show the highest average sea surface temperatures (Fig. 1b) as well as low daily temperature variation (Fig. S3) (see also paragraph 5 of the Results section in the revised manuscript).

The second factor has already been discussed more thoroughly in the original manuscript: nearly all tropical ocean locations show predominantly negative pre-rainfall temperature gradients (see Fig. S4). For bursts with a negative temperature gradient, we find a linear correlation between the magnitude of the temperature decline and the extreme rainfall intensity. This correlation leads to a separation (convergence) of the temperature curves of bursts in different intensity groups, as exemplarily shown in Fig. 2a (Fig. 2b).

Together, these two factors govern the temperature-rainfall scaling over tropical oceans. Low initial temperature separation (at $t=-24$ hours) and the negative correlation of rainfall intensity with temperature gradients leads to negative temperature-rainfall scaling (Figs. 2a, c and e). For locations with high initial temperature separation, the rainfall intensity-temperature gradient correlation actually reduces the strongly positive temperature-rainfall scaling, which would be even higher without the effect of this correlation (Figs. 2b, d and f).

To quantify the overlap between long-term average SSTs and α -values, we looked at the spatial correlation of these variables (Fig. S13). From 26°C upwards (approximately the temperature threshold for cyclogenesis), we obtain a negative correlation between long-term average SSTs and α -values. Negative α -values are heavily concentrated in regions with SSTs above approximately 28°C , corresponding to the contour line in Fig. 1a. This is also in line with the observation that the strongest temperature declines occur in those regions with the highest SSTs (see Fig. 3a). These findings help us to understand the following remaining questions: Why are most rainfall bursts over tropical oceans preceded by negative temperature gradients? And why are those pre-rainfall temperature gradients correlated with the subsequent rainfall intensity?

To answer these questions, we repeated the entire study using only bursts that we tagged as TC-related using the IBTrACS tropical cyclone archive. Generally, results look very similar for TC-tagged bursts compared to using all bursts. In the revised manuscript, the geographical map of temperature-rainfall scaling factors using all bursts is depicted in Fig. 1a, and using only TC-tagged bursts in Fig. S11. Although the spatial variation of α -values using only TC-tagged bursts is more noisy due to the reduced amount of data, the pattern of negative and positive temperature-rainfall scaling factors is in good agreement. Furthermore, the correlation between pre-rainfall temperature gradients and extreme rainfall intensities is nearly identical for the two cases (compare Fig. 4a and b). Although rainfall rates show higher values overall using only TC-tagged bursts, the Pearson correlation coefficient is almost the same (PCC= -0.999 for TC-tagged bursts, compared to PCC= -0.998 for all bursts), and so is the slope of the linear regression ($-39.7 \text{ mmh}^{-1}/^{\circ}\text{Ch}^{-1}$ for TC-tagged bursts, compared to $-39.9 \text{ mmh}^{-1}/^{\circ}\text{Ch}^{-1}$ for all bursts). Essentially, the only difference between the scaling using all bursts versus TC-tagged bursts is thus an offset on the y-axis.

For TCs, it has been demonstrated that

- L299ff.: TCs cause a substantial cooling of the upper layers of the ocean and the atmosphere along their tracks
- L307ff.: The cooling can begin up to 2-3 days ahead of the arrival of a cyclone, and the spatial extent of this pre-arrival cooling can reach more than 1.000 km.
- L312ff.: The amplitude and spatial extent of the cooling correlates strongly with the intensity of a cyclone.
- L316f.: The extreme rainfall rate is also strongly correlated with the intensity of a cyclone (this is a result from our revised study, using TC-tagged rainfall bursts to analyze the correlation between maximum sustained wind speeds of cyclones and the 90th percentile of rainfall rates, see Fig. S14).

We argue that cyclone-related cooling leads to the decline we observe in the temperature histories of rainfall bursts over tropical oceans (see Figs. 2a and b, Fig. 3a and Fig. S4). Furthermore, the correlation between cyclone intensity and cooling as well as rainfall intensities (Fig. S14) explains the scaling that we observe

between temperature gradients and rainfall intensities (Fig. 4a and b). L319ff.: “In other words, it is the TC intensity that determines the strength of the pre-rainfall cooling as well as the subsequent extreme rainfall intensity.”

Considering the relationships between TC intensity, TC-related pre-rainfall cooling and subsequent rainfall intensities, we interpret the similarity between the rainfall intensity-temperature gradient scaling for all bursts and TC-tagged bursts (Fig. 4a and b, respectively) as an indication that the predominant mechanism generating rainfall over oceans is cyclonic activity (see paragraph 8 in the Discussion section).

We have also added a figure showing the proportion of TC-tagged rainfall bursts among all bursts above the 90th percentile (see Fig. S10b). Up to 40-50% of bursts are TC-related over the northern tropical Atlantic, up to 90-100% over the northeastern pacific ocean, up to 70-80% over the northwestern pacific ocean, and up to 10-20% over the northern Indian ocean. When interpreting these numbers, it should be kept in mind that the list from the IBTrACS tropical cyclone archive is most likely not exhaustive. The strongest TCs are likely included there, but our arguments would also be valid for cyclonic activity that does not reach the thresholds to be defined as a TC.

With these results and explanations ironed out, we can finally explain the lack of overlap of TC tracks and negative scaling factors (L341ff.): Generally, one might expect a larger overlap of the regions with negative scaling (Fig. 1a) with the tropical cyclone tracks provided by IBTrACS (Fig. S10a), but a few factors have to be considered in that regard. First, TC tracks only show the propagation of the eye of TCs. The cooling effect, however, may extend up to 1,000 km away from the eye. For instance, even though there are no tracks south of the equator in the TC season of the northern hemisphere, the cooling may very well affect the (tropical) southern hemisphere as seen in Fig. 3a. Second, not all cyclonic activity is captured in the IBTrACS archive, which only contains tracks of TCs with maximum sustained winds of at least 50-60 km/h.

1.2 Quantitative analyses are needed. Instead of tracks, I would suggest using track density, which can better highlight the TC activity. It is also interesting to check if there is a significant anti-correlation between TC density and scaling. Intense storms tend to produce stronger cold wake, so it is interesting to find ways to incorporate TC intensity into the analysis. With the best track data, the authors can create a dataset including both rainfall and temperature without TC and compare this with the original one to see if you get stronger negative scaling.

We agree with the referee that we needed to strengthen our arguments with more quantitative analyses, and thank them for their valuable suggestions.

- using track density instead of tracks: we considered showing the track density (see figure below) in the revised manuscript, but decided against it when we found the strong overlap between long-term average SSTs and temperature-rainfall scaling factors (see the answer to the last question). We did, however, include a figure of the per-pixel proportion of TC-tagged rainfall bursts among all bursts above the 90th percentile of intensity values (Fig. S10b).

- anti-correlation between TC density and scaling: since the overlap between long-term average SSTs and α -values is much stronger, we investigated whether there is an anti-correlation between these variables instead (see also the answer to the last question). And indeed, from 26°C upwards - approximately the SST threshold for cyclogenesis - we obtain a strongly negative correlation between SSTs and α -values (Fig. S13).
- TC intensity analysis: based on the additional analysis regarding TC-tagged rainfall bursts we performed for the revised manuscript, we now argue that it is the TC intensity (in terms of maximum

sustained wind speeds) that determines both the amplitude of the pre-rainfall cooling, as well as the subsequent extreme rainfall intensity (see also the answer to the last question). In paragraph 5 and following of the Discussion section, we cite studies showing that the TC intensity correlates positively with the amplitude of the TC-related cooling. Our own analysis shows that the TC intensity correlates positively with the extreme rainfall intensity as well. These correlations explain the correlation we find - to our knowledge for the first time - between pre-rainfall temperature-gradients and extreme rainfall intensity (Fig. 4a and b).

- comparing scaling behaviour with and without TC-tagged rainfall bursts: in the revised manuscript, we compare temperature-rainfall scaling factors considering all bursts with scaling factors considering only TC-tagged bursts, which is similar to the comparison suggested by the referee. L224ff.: “Figure S11 depicts a geographical map of α -values over water bodies using only TC-tagged rainfall episodes. Deviations from the thermodynamically expected CC scaling tend to be even larger (in both negative and positive direction) compared to Fig. 1a. Overall, there is more scatter in the spatial variation of α -values, which should be expected due to the reduced amount of data. Nevertheless, the geographical patterns of negative and positive temperature-rainfall scaling factors are in good agreement.” Additionally, we compare the rainfall intensity-temperature gradient scaling for the two cases (all bursts versus only TC-tagged bursts, L236ff.). We observe a nearly identical rainfall intensity-temperature gradient correlation over northern tropical oceans (compare Fig. 4a with b). Although rainfall rates show higher values overall using only TC-tagged bursts, the Pearson correlation coefficient is almost the same (PCC=-0.999 for TC-tagged bursts, compared to PCC=-0.998 for all bursts), and so is the slope of the linear regression ($-39.7 \text{ mmh}^{-1}/^{\circ}\text{Ch}^{-1}$ for TC-tagged bursts, compared to $-39.9 \text{ mmh}^{-1}/^{\circ}\text{Ch}^{-1}$ for all bursts). Essentially, the only difference between the scaling using all bursts versus TC-tagged bursts is thus an offset on the y-axis. As mentioned in the answer to the last question, we interpret the similarity of the rainfall intensity-temperature gradient correlations for the two cases as an indication that the predominant mechanism generating rainfall over oceans is cyclonic activity.

1.3 The analyses here focus on pre-rainfall cooling. However, the cold wake of TCs typically occurs after the passage of the storm (e.g., Zhang et al., 2019). How to connect the two is another point concerning me.

It is true that we are focusing mainly on pre-rainfall cooling. By conducting the study considering only bursts without preceding rainfall events for at least 48 hours, we are effectively introducing a bias towards sampling rainfall events *before* the arrival of a cyclone, rather than during or after the passage of a cyclone. The reason we are selecting only bursts without preceding rainfall is that rainfall itself has a local cooling effect on the surface temperatures which influences the apparent temperature-rainfall scaling behaviour. We did, however, additionally perform the study using all rainfall bursts. In the revised manuscript, we have added a “robustness” paragraph (see paragraph 9 of the Results section), which includes a comparison of the rainfall intensity-temperature gradient correlation for all bursts vs bursts without preceding rainfall events (Figs. S6a and b). Although the slope of the regression between negative temperature gradients and extreme rainfall intensities is reduced by a factor of nearly two, the correlation is still perfectly linear (PCC=-0.997). We have also created a geographical map of α -values considering all bursts. In the figure below, we can see that the division between positive and negative scaling factors is in very good agreement.

Technically, we could show the results of our study using all rainfall bursts rather than only those without preceding rainfall. However, we believe that neglecting to take measures against the influence of rainfall on the apparent temperature-rainfall scaling would open the door to critique on the manner we set up the temperature-rainfall scaling analysis in our study.

2 The definition of bursts. The authors only consider ‘bursts without preceding rainfall for at least 48 hours’. Does it mean that only the onset three hours of rainfall in an extreme rainfall events with storm period longer than three hours is used as a sample? If so, many samples with extreme rainfall intensity would be excluded from the analysis. Please clarify.

We define bursts as consecutive 3-hourly rainfall events without interruption (see L389ff.). With each rainfall burst we associate a rainfall intensity, which we define as the maximum intensity of all rainfall events the burst is comprised of. So the answer to the referees question is technically no, we do not only use the onset three hours to define our samples. However, since each *burst* is considered a sample in our study, we end up with much less samples than we would if we considered all 3-hourly rainfall *events* as samples. We added the information on how we define the intensity of a burst in the Methods section (L392f.). In the original manuscript, this information was rather hidden in the caption of Fig. 2a (original manuscript). We hope that this change is sufficient to clarify how we collect samples.

Similar to the argument we stated in the answer to the last question, we accept the reduction in the number of samples for the sake of conducting the temperature-rainfall scaling analysis to the best of our knowledge. If we were to consider each 3-hourly rainfall event as a sample, and then take the subset of all samples without preceding rainfall, we would end up considering only the onset three hours of consecutive rainfall events. We believe, however, that the maximum intensity of consecutive rainfall events is a better representation of a burst’s intensity than the intensity of the onset 3-hourly rainfall event.

Note that in the aforementioned “robustness” paragraph we added in the revised manuscript, we additionally compare the rainfall intensity-temperature gradient correlation for two different definitions of the intensity of a burst: the *maximum* and the *mean* intensity of all rainfall events the burst is comprised of (Figs. S9a and b).

3 There are barely any references starting from the Result section. It is thus not clear how your results fit in previous work. Are they consistent or inconsistent with previous work? I think such information is a vital component of a manuscript, which can connect your work with the field.

We thank the referee for pointing out the need to better connect our results with existing studies. In the revised manuscript, we have added a substantial number of references in the Results and Discussion sections. Particularly the connection between TC-related atmospheric and oceanic dynamics and our results is much clearer now. By referencing appropriate articles and performing additional analyses using TC-tagged rainfall bursts, we are now in a position to argue that it is the TC intensity that determines the strength of the pre-rainfall cooling as well as subsequent rainfall intensities, which is perfectly in line with our finding that the strength of the pre-rainfall temperature decline positively correlates with the subsequent extreme rainfall intensity. We did not, however, find any references investigating the relationship between pre-rainfall cooling and rainfall intensities. To the best of our knowledge, our study is the first analyzing this relationship.

Less major comments

1 TRMM dataset tends to overestimate light rainfall while underestimate extreme rainfall. A study (Villarini et al., 2011) shows that the underestimation of TC rainfall in TRMM can be large compared to finer-scale radar-based rainfall datasets. The uncertainty of the TRMM dataset can have an impact on the scaling. I do not expect the authors to resolve this issue, but some discussion on it would be useful.

We thank the referee for this information and the reference, and have added a note in the “robustness” paragraph of the revised manuscript. L201ff.: “With regards to the general robustness of our results, it should also be noted that the TRMM rainfall product has been shown to underestimate extreme rainfall compared to finer-scale radar-based rainfall datasets [45]. Although this might affect the scaling analyses performed in this study, there is no better alternative to the TRMM product over open oceans at this time.”

2 Line 9. It should be temperature-extreme rainfall scaling.

Thank you for this careful observation. We have changed the wording in L9ff. accordingly.

3 Line 141. the intensities of the bursts determine the strength of the decline. Is the causal relation true?

Again, thank you for the careful observation. It is a bad choice of words, we do not show any causal relationship. In the revised manuscript, we have taken care not to imply any causality in the relationships we investigate, and instead only speak of correlations.

4 Line 240 and probably other places. ‘-6 to -2 hours before the onset of the burst’ is quite confusing. Should it be -6 to -2 hours from. . .

Thank you for pointing this out, indeed this was confusing and we have changed it to “6 to 2 hours before the onset of the burst”.

5 Figures. The caption of Fig. 2 and afterward should mention that the plots are based on the two boxes in Fig. 1b.

We thank the referee for the suggestion. In the revised manuscript, we added corresponding titles to the plots that are based on the two boxes in Fig. 1a, and also mention it in the captions (see Fig. 2 of the revised manuscript).

Reference

Zhang, J., Lin, Y., Chavas, D. R., & Mei, W. (2019). Tropical cyclone cold wake size and its applications to power dissipation and ocean heat uptake estimates. *Geophysical Research Letters*, 46, 10177– 10185. <https://doi.org/10.1029/2019GL083783> Villarini, G., Smith, J. A., Baeck, M. L., Marchok, T., and Vecchi, G. A. (2011), Characterization of rainfall distribution and flooding associated with U.S. landfalling tropical cyclones: Analyses of Hurricanes Frances, Ivan, and Jeanne (2004), *J. Geophys. Res.*, 116, D23116, doi:10.1029/2011JD016175.

REVIEWER COMMENTS

Reviewer #1 (Remarks to the Author):

The modified version addressed all the issues raised and the authors satisfactorily incorporated all the comments. So I recommend this modified version for publication.

The authors have done an excellent job in replying to my criticisms. However, I am very surprised by the fact that they have made in my opinion substantially more confusing (and possibly wrong) part of the interpretation of their results. In particular, they pay a lot of attention now to the degree of separation of pre-rainfall temperatures for regions with negative or positive scaling, using it sometimes as a cause or explanation (?) for the presence of negative scalings with temperature, which is honestly beyond my ability to comprehend.

I have here a list of points for the authors to address, mostly related to this problem. The rest of the work is excellent, so expect the paper to be ready for publication once the authors have clarified these parts.

Specific comments

16: specify pre-rainfall cooling of what (the readers have not yet read the paper)

133-141: a lot of text is used here and in other parts of the paper describing this effect, but I don't understand what is interesting or surprising about it. It's obvious that if you sample values from a location with larger/smaller variance you will observe larger/smaller differences among the values. Indeed I think that whether the spread of the pre-rainfall temperatures in a location is larger than in another should be commented after normalising by the local standard deviations, to remove the effect of the "intrinsic" variability of each location. But it all depends on the point you want to make, which again, is really not clear to me. If it's only justifying the larger/smaller spread in the different locations with the larger/smaller local standard deviations, this is a trivial observation and should be done in just one sentence;

139: beside my comment above, the term "saturation effect for temperatures" is definitely not appropriate for describing this phenomenon. There is no "saturation" of temperature in place at all here. The daily temperature in the tropical oceans has simply a smaller time variability than elsewhere, but this has nothing to do with a "saturation". Saturation means that something is steadily increasing (in time or as a function of another control variable or parameter), but with decreasing margins of increment, until it saturates to a plateau. I don't see how this concept applies to the present case, and I think that references to a "saturation effect" should be removed from the paper;

262-280: I completely miss the logic of the first argument. What has the smaller separation of pre-rainfall temperatures (that is, the smaller local variance of temperature) to do with having a negative scaling with temperature? Yes, they occur in the same regions, but the way this paragraph is set up, it seems that you are claiming that the smaller separation causes or explains the negative scaling, which is incomprehensible to me. This part needs a serious reworking. If you are claiming that the smaller separation causes the negative scaling, you have to provide an explanation why, that you don't provide at the moment. If you are not, you have to rewrite the paragraph because as it is now it seems that you are instead claiming this;

262-280: I am also puzzled by the conclusion of the second argument, where again you invoke this larger or smaller separation of pre-rainfall temperatures. Before that the argument is fine: what matters is the intensity of the cyclone, so higher rainfall is due to stronger cyclones which cause also larger temperature drops, and this results in higher rainfall being associated to lower temperature at the onset of the precipitation event, hence the negative scaling. But then what has the pre-rainfall temperature separation you mention again in 278-280 to do with all this?

275: "perfectly linear correlation" is an inappropriate term in my opinion. "Linear correlation" means something very precise, which is not what I think you want to say here. Maybe you meant "perfectly linear relation". In any case, the scalings you find are not "perfectly" linear in any mathematical sense. This is an overstatement that could be ok in an informal conversation, not in a scientific paper. Please rephrase this part;

283-285: "Low initial temperature separation and the negative correlation of rainfall intensity with temporal temperature gradients leads to negative temperature-rainfall scaling". Here you say it explicitly. I understand the second part of the argument, the first one is a total mystery to me. The smaller separation has no direct effect in causing a negative scaling. Given that the negative

scaling is due to the relation of extreme rainfall with a temperature drop due to the action of a cyclone, a larger initial temperature separation would simply give you a larger variability of the rainfall-temperature pairs. This would not result in a different value of the scaling parameter (and even less in a change of sign of it), but simply in a larger statistical uncertainty on its estimate;

321-324: I would be slightly more cautious in defining your results “the establishment of a quantitative relationship” without further considerations, for two reasons. First, the value of the scaling parameter depends quite a lot on the choice of the time range where the linear fit is applied. Second, while you provide a measure of the statistical significance of the regression, you don’t provide an estimate of the uncertainty on the value of the parameter (x plus or minus y). If you could provide this estimate I think that you could be justified in using this sentence. Otherwise, it would be appropriate to accompany it with a comment on the lack of the estimate of a statistical error at this stage of your work (plus a warning that choosing different time ranges for the linear regression leads to different values of the coefficient). Or just claim that you simply get the order of magnitude of the coefficient, which is already something.

Reviewer #3 (Remarks to the Author):

I want to first thank the authors for your great effort in new analyses and discussion. The current version looks much better. Yet, I still have a few concerns.

1 Lines 325-328. "Furthermore, the fact that the scaling between temporal temperature gradients and rainfall intensities for all rainfall episodes is essentially a shifted version of the scaling for TC-associated rainfall episodes indicates that the predominant mechanism generating rainfall over oceans is cyclonic activity." I still have some concerns regarding the claim of "essentially a shifted version". First I acknowledge that for regions with high TC activity TC does play an important role in the 90th rain events (Fig. S10 b). But there are extensive regions (e.g., near equator) where rainfall is not dominated by TCs. I agree with the authors that Ibtracs data do not include weak storms that I assume outnumbers TCs and may behave in a similar way to TCs regarding T-P scaling. But these storms tend to produce lighter rainfall than intense storms. And for near-equator oceans, non-rotation convection plays an important role in rainfall due to weak Coriolis force. A little more discussion on this would be useful.

2 Title. The work focuses on the tropical region, "global temperature-rainfall scaling" seems to not fit well.

3 Lines 319-321. It is a too strong claim that TC intensity determines the TC rainfall since there are many other factors.

4 Fig. S9 caption. rather then -> rather than.

5 Fig. S13. Maybe better to use contour for density rather than dots.

6 I have a more general question. What does the negative T-P scaling in TCs imply? Can it be consistent with the findings that global warming leads to increased TC (extreme) rainfall?

Responses to reviewer # 1

Reviewer #1 Remarks to the Author:

The modified version addressed all the issues raised and the authors satisfactorily incorporated all the comments. So I recommend this modified version for publication.

Thank you for recommending our paper for publication, we are glad that we could respond to all your concerns.

Responses to reviewer # 2

Reviewer #2 Remarks to the Author:

The authors have done an excellent job in replying to my criticisms. However, I am very surprised by the fact that they have made in my opinion substantially more confusing (and possibly wrong) part of the interpretation of their results. In particular, they pay a lot of attention now to the degree of separation of pre-rainfall temperatures for regions with negative or positive scaling, using it sometimes as a cause or explanation (?) for the presence of negative scalings with temperature, which is honestly beyond my ability to comprehend.

Thank you very much for your thorough and detailed assessment of our revised manuscript. In particular, we are grateful that you spotted a part of our explanations that was clearly more confusing and misleading than helpful for comprehension. Essentially, we mixed physical causes and heuristic elaborations in an unfortunate way. Since our explanations regarding the “initial temperature separation” are not necessary to understand our main findings, but rather distract from them, we decided to remove them from the revised manuscript. Instead, we have added two paragraphs in the discussion section that further explain the different physical aspects contributing to the empirical temperature-rainfall scaling (see second and third paragraph of the Discussion section, L189 - L207).

However, we would still like to explain the motivation behind our elaborations concerning the “initial temperature separation” in the last version of our manuscript, and answer all the questions raised by the referee on a point-by-point basis further below. First, we will provide a full explanation here, and then refer back to it when responding to your specific comments.

Essentially, there are two interacting physical principles relevant in the analysis of the scaling between temperatures and rainfall intensities: thermodynamics and circulation dynamics.

The most well-known thermodynamic aspect of the temperature-rainfall scaling is given by the Clausius-Clapeyron (CC) relation. It implies that – due to the increased water holding capacity of a warmer atmosphere – rainfall events should be stronger the higher the temperatures. This has been widely studied already, especially in the context of the impacts of anthropogenic global warming, but mostly over land and less comprehensively over the oceans. The CC effect leads to a positive contribution to the temperature-rainfall scaling.

In our study, we identify a dynamical mechanism that contributes substantially to the temperature-rainfall scaling over tropical oceans. The characteristics of cyclonic activity (i.e. wind-driven oceanic surface divergence and advective air transport) result in an intensity-dependent dynamical cooling of the near-surface temperature before the rainfall event. This leads to a negative contribution to the temperature-rainfall scaling.

These two contributions to the scaling are therefore in competition with each other, adding in opposing direction to the empirical scaling factors α that we estimate from the data. Besides, it should also be mentioned that there are without doubt additional (thermo)dynamic contributions to the scaling, given the very

strong positive scaling factors we observe over parts of the oceans.

Regarding the “initial temperature separation”, we found that locations with strongly negative scaling factors typically show less spread between the atmospheric temperatures of different intensity groups long before the rainfall events (24 hours) compared to locations with strongly positive scaling factors (compare the temperature histories in Figs. 2a and b). In the extreme (hypothetical!) case of assuming that only the above two contributions exist, and having no spread in the initial temperatures (i.e., the temperatures of the atmosphere at $t=-24\text{h}$, thus before the pre-rainfall cooling effect would become relevant), the scaling factor α , empirically measured at $t=-2\text{h}$, would be fully determined by the negative contribution of the pre-rainfall cooling, while there would be no measurable contribution from the CC-related (positive) effect. In this sense, the spread between initial temperatures influences the measured values of α . But, and here the referee is of course completely right, it was very confusing to mix this statistical effect with the physical explanation of the pre-rainfall cooling. We thus admit that we have not presented this line of argumentation well in the last version of our manuscript.

We now realize that explaining this in detail would require more space than the argument is actually worth. In essence, we were trying to provide further information as to why we observe the pattern of negative and positive scaling factors over tropical oceans, but without providing an actual physical explanation in addition to the pre-rainfall cooling, which is the relevant mechanism. Therefore, we decided to remove all parts related to the “initial temperature separation” from the revised manuscript. An additional reason we decided to discard our elaborations on the “initial temperature separation”, is that our main finding (the TC-related cooling mechanism) itself already explains the pattern (third paragraph of the Discussion section in the revised manuscript, L199ff.): “The influence of the intensity-dependent pre-rainfall cooling effect on the temperature-rainfall scaling grows with the proportion of episodes preceded by negative temporal temperature gradients. The strongest pre-rainfall temperature declines are observed in those regions that show a negative temperature-rainfall scaling (compare Fig. 1a and Fig. 3a). This indicates that the contribution of the pre-rainfall cooling effect dominates the scaling behaviour in those regions, leading to a net negative temperature-rainfall scaling.”

We hope that with these changes the manuscript is not just more succinct, but a lot clearer as well. Please find further responses to the specific points below.

I have here a list of points for the authors to address, mostly related to this problem. The rest of the work is excellent, so expect the paper to be ready of publication once the authors have clarified these parts.

Thank you for the acknowledgment, and your detailed list of concerns. Please find point-by-point responses, as well as indications of what we changed in the respective parts of the main text, below.

Specific comments

16: specify pre-rainfall cooling of what (the readers have not yet read the paper)

Thank you for the suggestion. We have changed this to “pre-rainfall cooling of near-surface air temperature”.

133-141: a lot of text is used here and in other parts of the paper describing this effect, but I don't understand what is interesting or surprising about it. It's obvious that if you sample values from a location with larger/smaller variance you will observe larger/smaller differences among the values. Indeed I think that whether the spread of the pre-rainfall temperatures in a location is larger than in another should be commented after normalising by the local standard deviations, to remove the effect of the “intrinsic” variability of each location. But it all depends on the point you want to make, which again, is really not clear to me. If it's only justifying the larger/smaller spread in the different locations with the larger/smaller local standard deviations, this is a trivial observation and should be done in just one sentence;

We apologize for this quite confusing paragraph, which we have removed in the revised manuscript. Even if we assumed that smaller/larger daily SST standard deviations result in smaller/larger temperature sep-

arations, we still had not provided any reasonable physical explanation as to why the standard deviations are smaller in regions with negative scaling (see next point). Apart from that, please also see our general explanation above.

139: beside my comment above, the term “saturation effect for temperatures” is definitely not appropriate for describing this phenomenon. There is no “saturation” of temperature in place at all here. The daily temperature in the tropical oceans has simply a smaller time variability than elsewhere, but this has nothing to do with a “saturation”. Saturation means that something is steadily increasing (in time or as a function of another control variable or parameter), but with decreasing margins of increment, until it saturates to a plateau. I don’t see how this concept applies to the present case, and I think that references to a “saturation effect” should be removed from the paper;

We fully agree that “saturation” was a very unfortunate term in this context, and have removed it from the manuscript. As the referee rightly states, there is no saturation effect that would explain the smaller standard deviations in regions with negative scaling factors.

262-280: I completely miss the logic of the first argument. What has the smaller separation of pre-rainfall temperatures (that is, the smaller local variance of temperature) to do with having a negative scaling with temperature? Yes, they occur in the same regions, but the way this paragraph is set up, it seems that you are claiming that the smaller separation causes or explains the negative scaling, which is incomprehensible to me. This part needs a serious reworking. If you are claiming that the smaller separation causes the negative scaling, you have to provide an explanation why, that you don’t provide at the moment. If you are not, you have to rewrite the paragraph because as it is now it seems that you are instead claiming this;

We apologize for the unclear presentation here, mixing a physical argument with a heuristic argument. The referee is right that the initial temperature separation does by no means cause the negative scaling. Instead, the negative scaling is caused by the physical effect of the intensity-dependent pre-rainfall cooling associated with cyclonic activity. We have rewritten this paragraph according to the general explanation above (see also paragraphs two and three of the Discussion section in the revised manuscript).

262-280: I am also puzzled by the conclusion of the second argument, where again you invoke this larger or smaller separation of pre-rainfall temperatures. Before that the argument is fine: what matter is the intensity of the cyclone, so higher rainfall is due to stronger cyclones which cause also larger temperature drops, and this results in higher rainfall being associated to lower temperature at the onset of the precipitation event, hence the negative scaling. But then what has the pre-rainfall temperature separation you mention again in 278-280 to do with all this?

Yes, we fully agree that the intensity of the cyclone determines the strength of the pre-rainfall cooling as well as the rainfall intensity. The pre-rainfall temperature separation has nothing to do with this effect and does not contribute to it. Please also refer to our general explanation above and the response to your last point.

275: “perfectly linear correlation” is an inappropriate term in my opinion. “Linear correlation” means something very precise, which is not what I think you want to say here. Maybe you meant “perfectly linear relation”. In any case, the scalings you find are not “perfectly” linear in any mathematical sense. This is an overstatement that could be ok in an informal conversation, not in a scientific paper. Please rephrase this part;

We agree, and changed this to “strong linear correlation”.

283-285: “Low initial temperature separation and the negative correlation of rainfall intensity with temporal temperature gradients leads to negative temperature-rainfall scaling“. Here you say it explicitly. I understand the second part of the argument, the first one is a total mystery to me. The smaller separation has no direct effect in causing a negative scaling. Given that the negative scaling is due to the relation of extreme rainfall with a temperature drop due to the action of a cyclone, a larger initial temperature separation would simply give you a larger variability of the rainfall-temperature pairs. This would not result in a different

value of the scaling parameter (and even less in a change of sign of it), but simply in a larger statistical uncertainty on its estimate;

Yes, you are right, this is a good example of how we unfortunately mixed the physical explanation of the intensity-dependent pre-rainfall cooling and the heuristic argument about the “initial temperature separation”. As mentioned in the general explanation above, we removed the heuristic argumentation from the revised manuscript.

321-324: I would be slightly more cautious in defining your results “the establishment of a quantitative relationship” without further considerations, for two reasons. First, the value of the scaling parameter depends quite a lot on the choice of the time range where the linear fit is applied. Second, while you provide a measure of the statistical significance of the regression, you don’t provide an estimate of the uncertainty on the value of the parameter (x plus or minus y). If you could provide this estimate I think that you could be justified in using this sentence. Otherwise, it would be appropriate to accompany it with a comment on the lack of the estimate of a statistical error at this stage of your work (plus a warning that choosing different time ranges for the linear regression leads to different values of the coefficient). Or just claim that you simply get the order of magnitude of the coefficient, which is already something.

Following your comment that we do not provide uncertainties on the scaling factors, we added a comprehensive uncertainty quantification, including error bars for each 90th percentile bin and uncertainties for the slopes (see Figs. 2c, d, e, f and Figs. 4a and b, as well as the Methods section, L340ff.). We did, however, tone down our statement a little (L229): “To our knowledge, this relationship has not been quantified before.”.

Responses to reviewer # 3

Reviewer #3 Remarks to the Author:

I want to first thank the authors for your great effort in new analyses and discussion. The current version looks much better. Yet, I still have a few concerns.

We want to thank the reviewer for their acknowledgment. Please find our point-by-point responses to your concerns below.

1 Lines 325-328. “Furthermore, the fact that the scaling between temporal temperature gradients and rainfall intensities for all rainfall episodes is essentially a shifted version of the scaling for TC-associated rainfall episodes indicates that the predominant mechanism generating rainfall over oceans is cyclonic activity.” I still have some concerns regarding the claim of “essentially a shifted version”. First I acknowledge that for regions with high TC activity TC does play an important role in the 90th rain events (Fig. S10 b). But there are extensive regions (e.g., near equator) where rainfall is not dominated by TCs. I agree with the authors that Ibracs data do not include weak storms that I assume outnumbers TCs and may behave in a similar way to TCs regarding T-P scaling. But these storms tend to produce lighter rainfall than intense storms. And for near-equator oceans, non-rotation convection plays an important role in rainfall due to weak Coriolis force. A little more discussion on this would be useful.

Thank you for this very valid point. We believe the statement itself is justified. The scaling in Fig. 4a is virtually a shifted version of the scaling in Fig. 4b. The range of temperature gradient values, the slope of the regression, and the Pearson correlation coefficient are very similar, the only difference is a shift on the y-axis. The interpretation of that observation, however, could clearly benefit from some more discussion, as the referee rightly notes. Therefore, we added a few sentences to the corresponding paragraph (L230ff.): “Furthermore, the fact that the scaling between temporal temperature gradients and rainfall intensities for all rainfall episodes is essentially a shifted version of the scaling for TC-associated rainfall episodes indicates that the predominant mechanism generating rainfall over oceans is cyclonic activity. There are, of course, other mechanisms that generate rainfall over tropical oceans. For instance, convective rainfall systems near the equator that are non-rotating because of the lack of a sufficiently strong Coriolis force, produce strong

rainfall. Two possible reasons that these mechanisms do not noticeably alter the shape of the scaling in Fig. 4a compared to Fig. 4b are: they are statistically outweighed by cyclonic activity; or they are associated with positive temporal pre-rainfall temperature gradients. Further analysis, however, would be required to better understand their influence on the correlation between temperature gradients and rainfall intensities.”

2 Title. The work focuses on the tropical region, “global temperature-rainfall scaling” seems to not fit well.

We agree with the referee, and changed the title to: “The role of cyclonic activity in tropical temperature-rainfall scaling”.

3 Lines 319-321. It is a too strong claim that TC intensity determines the TC rainfall since there are many other factors.

We agree with the reviewer, and removed this sentence from the revised manuscript, which helped making the paragraph more succinct as well.

4 Fig. S9 caption. rather then → rather than.

Thank you for the pointer, we corrected the typo accordingly.

5 Fig. S13. Maybe better to use contour for density rather than dots.

We thank the referee for the suggestion. A density plot has been added to the corresponding figure.

6 I have a more general question. What does the negative T-P scaling in TCs imply? Can it be consistent with the findings that global warming leads to increased TC (extreme) rainfall?

Yes, it’s fully consistent with the thermodynamic Clausius-Clapeyron relationship, via which – due to the increased water-holding capacity of a warmer atmosphere – rainfall events should be expected to become stronger because of anthropogenic global warming. One of the key points of our study is that, when attempting to quantify how strongly rainfall events increase in magnitude per degree of warming directly from observations, the pre-rainfall cooling needs to be recognized and taken into account in order to avoid a negatively biased estimate of the scaling. We have added two sentences to the manuscript clarifying that our results do not contradict previous studies finding that global warming leads to increased rainfall extremes (L256ff.): “Hence, on average, the intensity of extreme rainfall events will increase with rising atmospheric temperatures, which is consistent with the large number of studies on the effect of global warming on rainfall extremes. However, this thermodynamic effect is accompanied by pronounced dynamical effects that lead to complex spatial patterns.”

REVIEWERS' COMMENTS

Reviewer #2 (Remarks to the Author):

I thank the authors for addressing my criticisms and suggestions. The revised version of the paper is in my opinion ready for publication. Congratulations for the very nice work!

Francesco Ragone
Université Catholique de Louvain
Royal Meteorological Institute of Belgium

Reviewer #3 (Remarks to the Author):

The authors have well addressed all my concerns, and I am happy to recommend publication.